EMBO
Molecular Medicine

# Crosstalk between CD64+MHCII+ macrophages and CD4+ T cells drives joint pathology during chikungunya

Fok-Moon Lum [1,11 ✉], Yi-Hao Chan[1,11], Teck-Hui Teo[1], Etienne Becht[2,3], Siti Naqiah Amrun[1], Karen WW Teng[2], Siddesh V Hartimath[4], Nicholas KW Yeo [1], Wearn-Xin Yee[2], Nicholas Ang[2], Anthony M Torres-Ruesta[1], Siew-Wai Fong[1], Julian L Goggi [4], Evan W Newell[2,3], Laurent Renia [1,2,5,6], Guillaume Carissimo [1,7] & Lisa FP Ng [1,8,9,10 ✉]

## Abstract

**Communications between immune cells are essential to ensure appropriate coordination of their activities. Here, we observed the infiltration of activated macrophages into the joint-footpads of chikungunya virus (CHIKV)-infected animals. Large numbers of CD64+MHCII+ and CD64+MHCII- macrophages were present in the joint-footpad, preceded by the recruitment of their CD11b+Ly6C+ inflammatory monocyte precursors. Recruitment and differentiation of these myeloid subsets were dependent on CD4+ T cells and GM-CSF. Transcriptomic and gene ontology analyses of CD64+MHCII+ and CD64+MHCII- macrophages revealed 89 differentially expressed genes, including genes involved in T cell proliferation and differentiation pathways. Depletion of phagocytes, including CD64+MHCII+ macrophages, from CHIKV-infected mice reduced disease pathology, demonstrating that these cells play a pro-inflammatory role in CHIKV infection. Together, these results highlight the synergistic dynamics of immune cell crosstalk in driving CHIKV immunopathogenesis. This study provides new insights in the disease mechanism and offers opportunities for development of novel anti-CHIKV therapeutics.**

**Keywords** Chikungunya; Immune Crosstalk; Immunopathogenesis GM-CSF
**Subject Categories** Immunology; Microbiology, Virology & Host Pathogen Interaction

## Introduction

Chikungunya is a neglected tropical disease caused by infection with the chikungunya virus (CHIKV), an alphavirus transmitted by mosquitoes (mainly *Aedes aegypti* and *Aedes albopictus*) (Powers et al, 2000). This disease is typically self-limiting and characterized by common symptoms such as myalgia, fever and rash (Dupuis-Maguiraga et al, 2012). However, it is capable of causing devastating long-term effects in rare cases, particularly in newborns and the immunocompromised (Kee et al, 2010; van Ewijk et al, 2021). In particular, it is important to identify the specific immune cell populations involved in CHIKV infection, clarifying the mechanisms of host-virus interaction and opening new avenues for immunological therapeutics against chikungunya.

Long-term joint pain is a particularly common and debilitating complication of chikungunya and earlier studies showed that CD4+ T cells drive joint-footpad inflammation in CHIKV-infected mouse models (Teo et al, 2013). These studies also led to the recommendation of fingolimod (Teo et al, 2017) and abatacept (Miner et al, 2017) as potential therapeutic agents that could block pathogenic CD4+ T cells from infiltrating into the infected joint-footpad during peak CHIKV pathology. However, it is likely that these CD4+ T cells act in tandem with other myeloid cell (Roberts et al, 2015), and drugs targeting CD4+ T cells may therefore fail to address the full immunology that drives joint pathology in CHIKV. Monocytes and monocyte-derived cells contribute to both protection and pathology during CHIKV infection (Carissimo et al, 2019; Felipe et al, 2020; Haist et al, 2017; Rulli et al, 2011). Given the functional diversity of myeloid cells in regulating host immune responses, it is important to identify and characterize the unique myeloid subsets that impact CHIKV pathology, thereby enhancing our understanding of host-virus interactions in the context of CHIKV. Moreover, it will also bring valuable knowledge on

[1] A*STAR Infectious Diseases Labs (A*STAR ID Labs), Agency for Science, Technology and Research, Singapore 138648, Singapore. [2] Singapore Immunology Network (SIgN), Agency for Science, Technology and Research, Singapore 138648, Singapore. [3] Vaccine and Infectious Disease Division, Fred Hutchinson Cancer Research Center, Seattle, WA 98109, USA. [4] Institute of Bioengineering and Bioimaging (IBB), Agency for Science, Technology and Research, Singapore 138648, Singapore. [5] Lee Kong Chian School of Medicine, Nanyang Technological University, Singapore 636921, Singapore. [6] School of Biological Sciences, Nanyang Technological University, Singapore 637551, Singapore. [7] Infectious Diseases Translational Research Programme, Yong Loo Lin School of Medicine, National University of Singapore, Singapore 117545, Singapore. [8] Department of Biochemistry, Yong Loo Lin School of Medicine, National University of Singapore, Singapore 117597, Singapore. [9] National Institute of Health Research, Health Protection Research Unit in Emerging and Zoonotic Infections, University of Liverpool, Liverpool L69 7BE, UK. [10] Institute of Infection, Veterinary and Ecological Sciences, University of Liverpool, Liverpool L69 7ZX, UK. [11] These authors contributed equally: Fok-Moon Lum, Yi-Hao Chan. ✉E-mail: lum_fok_moon@idlabs.a-star.edu.sg; lisa_ng@idlabs.a-star.edu.sg

immune cell crosstalk during CHIKV infection and possibly identify new therapeutic targets.

First, we attempted to identify the specific macrophage population involved in mediating CD4+ T cell responses. Subsequently, we were interested in elucidating the immune dynamics between these cell populations and identified key immune components in driving CHIKV pathogenesis. In this study, we used CyTOF (Cytometry by Time-Of-Flight) to first identify two unique populations of CD64+ macrophages that infiltrate the joint-footpad of CHIKV-infected animals. We next investigated their potential crosstalk with joint-infiltrating CD4+ T cells in a series of in vivo and ex vivo assays. T cells were found to affect the recruitment and differentiation of the CD64+ macrophages via cytokine secretion. Transcriptomic profiling and in vivo phagocytes depletion just before the disease peak, indirectly demonstrated involvement of the CD64+ macrophages in priming CD4+ T cell activity, suggesting a functional crosstalk between these key immune subsets during peak CHIKV-induced pathology. Together, these data provide insights into the mechanisms of CHIKV immunopathogenesis and open new avenues for the development of anti-CHIKV immunotherapeutics targeting the pro-inflammatory macrophages.

# Results

## Mass cytometry reveals infiltration and activation of macrophages in the inflamed joint-footpad of CHIKV-infected mice

Previous studies have shown the involvement of CD4+ T cells and myeloid cells in CHIKV pathogenesis (Carissimo et al, 2019; Felipe et al, 2020; Haist et al, 2017; Lee et al, 2015; Miner et al, 2017; Rulli et al, 2011; Teo et al, 2017; Teo et al, 2015; Teo et al, 2013). To demonstrate the presence of myeloid cells in the joint-footpad, we injected CHIKV-infected mice with [18F]FEPPA, a radioligand for the 18-kDa translocator protein (TSPO) (Colasanti et al, 2014; Dedeurwaerdere et al, 2012; Goggi et al, 2021; Hatori et al, 2012; Missault et al, 2019; Vignal et al, 2018). This allowed the tracking of activated macrophages infiltrating into the inoculated joint-footpad (Fig. 1A). PET/CT images were taken at both 3 and 6 days post-infection (dpi), and the presence of activated macrophages at 6 dpi was observed to be at least two-fold higher in the virus-inoculated joint-footpad (Fig. 1B).

To further identify and characterize specific cellular changes within the CD11b+ myeloid cells infiltrating the joint-footpad, we performed CyTOF on joint-footpad cells harvested at 6 dpi, with an antibody panel specific to myeloid cell markers (Becher et al, 2014). Data obtained were subsequently visualized with the dimension reduction tool UMAP (Uniform Manifold Approximation and Projection) (Becht et al, 2019). The UMAP plots were normalized to 5000 cells per sample, and represent the proportion, but not the quantity of cells present in the joints. Nevertheless, UMAP analysis revealed substantial differences in the immune cell populations proportion in the joint-footpad of CHIKV-infected mice at the peak of pathology (6 dpi) compared to controls (Fig. 1C). Notably, we observed an increased proportion of clusters 7, 8, and 12 (Fig. 1C,D), which express high levels of surface markers CD11b and Ly6C (Fig. EV1A), identifying them as tissue-infiltrating inflammatory monocytes (Carissimo et al, 2019; Haist et al, 2017). Particularly, cluster 7 was of interest given that it also expressed high levels of CD64 (Fig. EV1A), a marker associated with activated monocyte-derived macrophages (Hristodorov et al, 2015;

Tamoutounour et al, 2012). Finally, we performed immunophenotyping on the infiltrating immune cells at 6 dpi (Fig. EV1B). Gating on the CD45+CD11b+Ly6C+Ly6G- inflammatory monocytes revealed that CD64 expression was upregulated following CHIKV infection (Fig. 1E). This CD64+ population could be further divided into CD64+MHCII+ and CD64+MHCII- subpopulations (Fig. 1E). Manual gating with the CyTOF data produced the same populations (Fig. 1F), complementing the immunophenotyping results. Importantly, this indicates the elevated presence of monocytes-derived macrophages in the joint-footpad, further corroborating with the data obtained with PET/CT imaging (Fig. 1A,B). However, it is important to note that TSPO can also be expressed by fibroblast-like synoviocytes (FLS) (Narayan et al, 2018), which play a pathogenic role in rheumatoid arthritis (Bartok and Firestein, 2010) Notably, increased proportion in clusters 7, 8, and 12 reflected a corresponding decreased proportion of clusters 4, 5, 6, and 9 (Fig. 1C,D). These clusters were identified as CD11blo innate lymphoid cells or CD11chi DCs (Figs. 1C and EV1A), and thus will not be the focus of this report.

## CD4+ T cells are key producers of IFNγ and GM-CSF during CHIKV infection

Factors that influence infiltration of CD11b+Ly6C+ inflammatory monocytes and their differentiation into CD64+MHCII- and CD64+MHCII+ subsets during CHIKV infection remain poorly described. IFNγ and GM-CSF, two cytokines produced by activated CD4+ T cells (Campbell et al, 2011), were recently shown to mediate the transition of inflammatory Ly6C+ monocytes into macrophages (Amorim et al, 2022). Given the importance of CD4+ T cells in the pathology of CHIKV (Carissimo et al, 2019; Miner et al, 2017; Teo et al, 2017; Teo et al, 2015; Teo et al, 2013), we investigated how these cells influence the phenotype of infiltrating inflammatory monocytes during CHIKV infection.

To determine whether CD4+ T cells present in CHIKV-infected joint-footpad produce IFNγ and GM-CSF, we isolated joint-footpad cells at 6 dpi and stimulated them with the T cell-activating agents phorbol myristate acetate (PMA) and ionomycin. A majority of the CD4+CD44+ memory/effector T cells (Fig. EV2A) produced IFNγ upon stimulation, and ~22% of these IFNγ-producing T cells were also capable of producing GM-CSF (Fig. EV2B and C).

## IFNγ and GM-CSF trigger differentiation of CD64+MHCII- cells into CD64+MHCII+ cells in vitro

To further explore the roles of IFNγ and GM-CSF in the differentiation of CD64+MHCII- and CD64+MHCII+ macrophages, we isolated peripheral blood immune cells from non-infected control animals for a differentiation assay. The majority of freshly isolated CD11b+Ly6C+ monocytes did not express CD64, and only a minor proportion (~14%) expressed MHCII (Fig. EV2D). We stimulated the isolated peripheral blood immune cells with 25 ng/mL of IFNγ, GM-CSF, or both. More than 90% of non-stimulated CD11b+Ly6C+ cells expressed CD64, while only ~25% expressed MHCII; this suggests that ex vivo culture conditions induce CD64 expression (Fig. EV2E). Nevertheless, a 24 h stimulation with IFNγ and/or GM-CSF increased the number of MHCII expressing cells by at least 2.5-fold (Fig. EV2E, panels 2–4), raising the proportion of CD64+MHCII+ cells within the CD11b+Ly6C+ monocytes from ~16% (non-stimulated) to 54% (stimulated with both IFNγ and

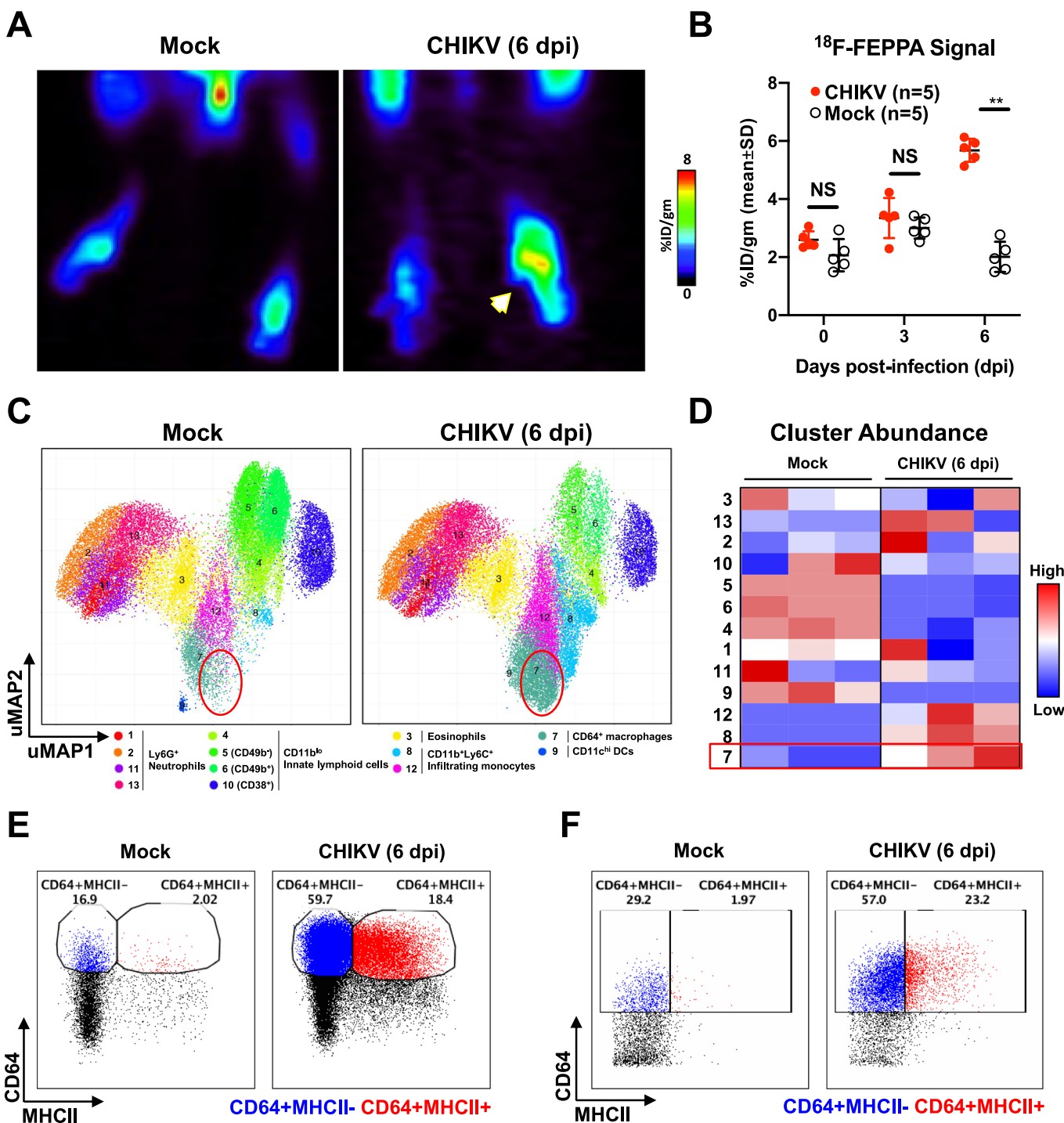

**A**

Mock | CHIKV (6 dpi)

%ID/gm

**B**
$^{18}$F-FEPPA Signal

● CHIKV (n=5)
○ Mock (n=5)

%ID/gm (mean±SD)

NS | NS | **

Days post-infection (dpi)

**C**

Mock | CHIKV (6 dpi)

uMAP2 / uMAP1

1 | 2 | 11 | 13 — Ly6G+ Neutrophils
4 | 5 (CD49b⁺) | 6 (CD49b⁺) | 10 (CD38⁺) — CD11b^lo Innate lymphoid cells
3 — Eosinophils | 8 — CD11b⁺Ly6C⁺ | 12 — Infiltrating monocytes
7 — CD64⁺ macrophages | 9 — CD11c^hi DCs

**D**
Cluster Abundance

Mock | CHIKV (6 dpi)

3 | 13 | 2 | 10 | 5 | 6 | 4 | 1 | 11 | 9 | 12 | 8 | 7

High / Low

**E**

Mock | CHIKV (6 dpi)

CD64+MHCII−
16.9 | CD64+MHCII+
2.02 | CD64+MHCII−
59.7 | CD64+MHCII+
18.4

CD64 / MHCII

**CD64+MHCII− CD64+MHCII+**

**F**

Mock | CHIKV (6 dpi)

CD64+MHCII−
29.2 | CD64+MHCII+
1.97 | CD64+MHCII−
57.0 | CD64+MHCII+
23.2

CD64 / MHCII

**CD64+MHCII− CD64+MHCII+**

GM-CSF) (Fig. 2A). Stimulation with GM-CSF and/or IFNγ increased MHCII expression by similar amounts (Figs. EV2E and 2B), but CD64 expression only increased in IFNγ-stimulated cells (Figs. EV2E and 2C).

## Depleting GM-CSF reduces the severity of chikungunya

Since both IFNγ and GM-CSF affected the phenotype of CD64⁺ macrophages in vitro, it was important to study their roles in vivo

during CHIKV infection. First, we treated CHIKV-infected wild-type mice at 4 dpi with anti-GM-CSF neutralizing antibodies and observed a reduction in the severity of joint-footpad swelling between 5–8 dpi compared to non-treated mice (Fig. 3A), with no effect on viral RNA load clearance (Fig. 3B). Neutralization of GM-CSF also reduced the presence of several immune populations (Fig. EV3), including the all-important CD4⁺ T cells and CD11b⁺Ly6C⁺ cells (Fig. 3C). However, in vivo GM-CSF depletion did not affect the differentiation of CD11b⁺Ly6C⁺ monocytes into

◄   **Figure 1.   Upregulation of CD64⁺ macrophages during CHIKV infection identified by high-dimensional analysis of mass cytometry data.**

(A) Representative PET/CT images of mock-infected and CHIKV-infected joints at 6 days post-infection (dpi) following intravenous injection of ¹⁸F-FEPPA radioligand to track infiltration of activated macrophages. Amount of [¹⁸F]FEPPA uptake was measured in the paw area and represented as a percentage of the administered dose per gram of tissue in the ROI (%ID/gm). White arrow indicates CHIKV-infected joint. (B) Radioactive signal from mock and CHIKV-infected joint-footpads was quantified. Data were from biological replicates and presented as mean ± SD. Statistical analysis was performed using non-parametric Mann–Whitney *U* test (two-tailed; **P = 0.0079). (C,D) Joint-footpad cells from CHIKV-infected and non-infected animals (*n* = 3 per group) were harvested at 6 dpi and stained with a panel of antibodies targeting myeloid cell surface markers. Acquisition was performed with CyTOF and data were analyzed with dimension reduction technique Uniform Manifold Approximation and Projection (UMAP). Superimposed PhenoGraphs of UMAP transformed CyTOF data from mock and CHIKV-infected joints. Presence of cluster 7 as enclosed by red circle (C). Cluster ID annotation with heatmap. Cluster IDs are indicated in rows, while groups are indicated in the columns. The color represents the Z-score transformed median number of cells in the clusters, which are grouped in phenotypic proximity based on surface marker similarities. Red rectangle highlights the difference of cluster 7 between CHIKV and mock-infected joints (D). (E,F) Manual gating of CD64⁺MHCII⁻ (blue) and CD64⁺MHCII⁺ (red) macrophages on live CD45⁺CD11b⁺Ly6C⁺Ly6G⁻ monocytes using data obtained from either fluorescence-based flow cytometry (E) or CyTOF (F). Source data are available online for this figure.

CD64⁺MHCII⁺ macrophages (Fig. 3D). Nevertheless, GM-CSF depletion reduced the overall numbers of tumor necrosis factor (TNF) α-, IFNγ-, Granzyme A-, Granzyme B-, and GM-CSF-secreting CD4⁺CD44⁺ T cells (Fig. 3E–I). Taken together, these observations suggest that GM-CSF contribute to CHIKV immunopathogenesis via the recruitment of immune cells into the virus-infected tissues.

Next, we investigated the role of IFNγ during CHIKV infection. We observed that CHIKV-induced joint swelling was mildly reduced (at 3 and 7–10 dpi) in IFNγ-deficient animals compared to wild-type mice (Fig. EV4A), with no effect on viral RNA load (Fig. EV4B). We also attempted to deplete IFNγ from CHIKV-infected wild-type animals and saw no differences in both disease severity and virus clearance (Figs. EV4C and D). These observations corroborated with the data from our previous publication (Teo et al, 2013). Immunophenotyping of joint-footpad-infiltrating immune cells (Fig. EV4E) at 6 dpi further revealed that IFNγ-deficiency did not affect the numbers of CD4⁺ T cells and CD11b⁺Ly6C⁺ cells infiltrating into the footpad (Fig. EV4F), but rather it significantly reduced the conversion of CD11b⁺Ly6C⁺ cells into CD64⁺MHCII⁺ macrophages by up to 10% (Fig. EV4G). This would suggests a minor but significant role of IFNγ possibly working in tandem with other cytokines to drive the macrophage conversion.

## CD4⁺ T cells regulate CD64⁺MHCII⁺ macrophages during CHIKV infection

Based on our results thus far, we hypothesise that CD4⁺ T cells may regulate the levels of CD64⁺MHCII⁺ macrophages in CHIKV-infected joints. To decipher this, we depleted CD4⁺ T cells from CHIKV-infected mice (Fig. EV5A). This reduced the CHIKV-induced joint-footpad swelling (Fig. EV5B), with no effect on viral RNA load clearance (Fig. EV5C) (Carissimo et al, 2019; Miner et al, 2017; Teo et al, 2017; Teo et al, 2013). Immunophenotyping showed that several immune populations infiltrating the virus-infected joint-footpad, including CD11b⁺Ly6C⁺ inflammatory monocytes, were significantly reduced in the absence of CD4⁺ T cells (Figs. EV5D and 4A). The percentage of CD11b⁺Ly6C⁺ inflammatory monocytes differentiating into CD64⁺MHCII⁺ macrophages was also significantly reduced from ~42% to 19% (Fig. 4B).

To further validate the function of CD4⁺ T cells in the differentiation of infiltrated monocytes, we quantified the levels of key immune mediators present in the joint-footpad of CHIKV-infected animals at 6 dpi. Twenty-six chemokines, growth factors, pro- and anti-inflammatory cytokines, including IP-10, MCP-1,

IL-12p70, IL-1β, TNFα, GM-CSF, and IFNγ, were elevated following CHIKV infection (Fig. 4C). However the levels of these immune mediators were reduced in CD4-depleted animals (Fig. 4C). The levels of GM-CSF and IFNγ, in particular, were significantly reduced by >90% (Fig. 4D). These results further suggest the importance of GM-CSF and IFNγ in the recruitment of CD11b⁺Ly6C⁺ inflammatory monocytes and their subsequent transition into CD64⁺MHCII⁺ macrophages, respectively.

## Crosstalk between CD64⁺MHCII⁺ macrophages and CD4⁺ T cells drives CHIKV pathology

Having shown that CD4⁺ T cells affect the recruitment and differentiation of CD11b⁺Ly6C⁺ inflammatory monocytes, we next investigated the functional roles of CD64⁺MHCII⁻ and CD64⁺MHCII⁺ macrophages during CHIKV infection. Bulk RNAseq analyses of CD64⁺MHCII⁺ and CD64⁺MHCII⁻ macrophages isolated from the joint-footpad of CHIKV-infected mice at 6 dpi revealed 89 differentially expressed genes (DEGs) with a false discovery rate (FDR) < 0.05 (Fig. 5A). Subsequent gene ontology (GO) analysis associated a total of 28 DEGs with 18 representative GO terms and pathways (Fig. 5B and Dataset EV1) following our selection criteria (see Methods). When the 18 GO terms are presented as nodes and clustered based on similarity of the genes present in each of the associated terms or pathways, five DEGs—*IL12B*, *CCR7*, *CD74*, *TBX21*, and *SERPINB9*—were associated with numerous GO terms and pathways (Fig. 5C and Dataset EV1). Nevertheless, the identified GO terms and pathways could be grouped into eight main functional groups (Fig. 5D and Dataset EV1), including T-helper cell differentiation and antigen processing and presentation. This was expected, since CD64⁺MHCII⁺ macrophages act as antigen-presenting cells to CD4⁺ T cells (Fig. 6A,B). However, it is important to note that CD64⁺ macrophages also play other roles during CHIKV pathogenesis, including regulating neutrophil migration and the apoptotic and cytotoxic activities of immune cells (Fig. 5D).

Finally, to illustrate a possible functional relationship between CD64⁺ macrophages and CD4⁺ T cells during active CHIKV infection, we depleted phagocytic cells, which includes macrophages, with clodronate liposomes at 5 dpi, a day before the peak of joint-footpad inflammation. This significantly decreased joint-footpad swelling from 6–10 dpi (Fig. 6C). Immunophenotyping of joint-footpad cells at 6 dpi showed that clodronate liposome treatment significantly reduced the presence of CD11b⁺Ly6C⁺ precursor cells, and both CD64⁺MHCII⁻ and CD64⁺MHCII⁺ macrophages (Fig. 6D). Clodronate liposome treatment also significantly reduced the numbers of CD4⁺ T cells (Fig. 6D).

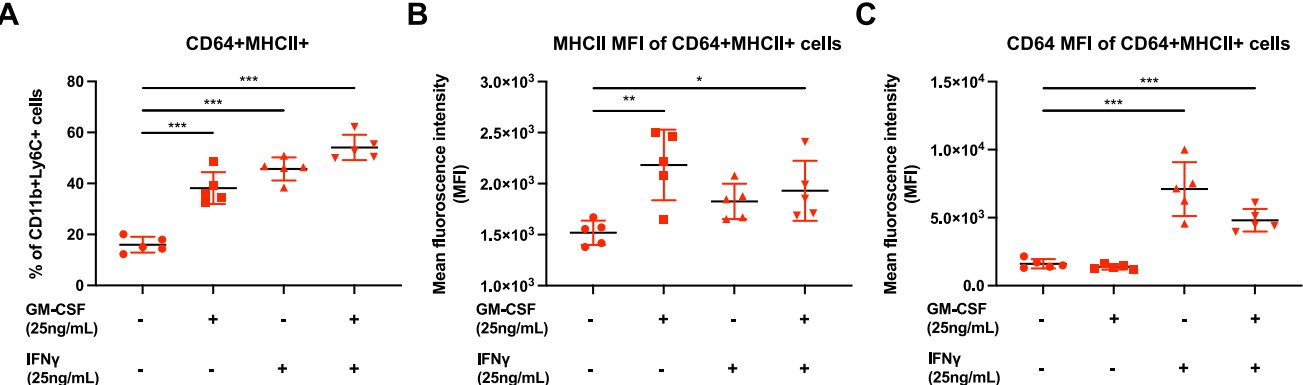

**Figure 2. Stimulation of peripheral whole blood with GM-CSF and/or IFNγ promotes the differentiation of CD64+MHCII- into CD64+MHCII+ macrophages.**

(A) Peripheral whole blood was obtained from non-infected animals ($n = 5$) and were subjected to GM-CSF and IFNγ stimulation after removal of the red blood cells. Stimulated cells were harvested 24 h later and stained for the identification of monocytes and macrophage subsets. Graphs illustrating the differentiation percentage of CD11b + Ly6C+ cells into CD64+MHCII+ macrophages. Data were from biological replicates and presented as mean ± SD. Data comparisons between the various groups were performed with one-way ANOVA with Dunnett's post test comparing against the control untreated group. Untreated vs GM-CSF, ***mean diff = −22.19; untreated vs IFNγ, ***mean diff = −29.74; untreated vs GM-CSF + IFNγ, ***mean diff = −38.13. (B,C) Plots showing the mean fluorescence intensity (MFI) of MHCII (B) and CD64 (C) staining on the CD64+MHCII+ macrophages. Data were from biological replicate and presented as mean ± SD. Data comparisons between the various groups were performed with one-way ANOVA with Dunnett's post test comparing against the control untreated group. For MHCII MFI, untreated vs GM-CSF, **mean diff = −664.2; untreated vs GM-CSF + IFNγ, *mean diff = −411.2. For CD64 MFI, untreated vs IFNγ, ***mean diff = −5490; untreated vs GM-CSF + IFNγ, ***mean diff = −3201. Source data are available online for this figure.

Interestingly, levels of IP-10, a key T-cell chemokine in arthritogenic alphaviruses (Lin et al, 2020), is not affected with clodronate liposomes treatment (Fig. 6E), suggesting that CD4+ T cells infiltration into the joint-footpad is not affected. Taken together, these results further suggest a potential immune crosstalk cycle between CD64+ macrophages and CD4+ T cells, wherein constant priming of cytotoxic CD4+ T cells, by CD64+MHCII+ macrophages, leads to their secretion of both GM-CSF and IFNγ, eventually leading to a vicious cycle of recruitment and activation of more CD64+MHCII+ macrophages. The elevated presence of CD64+MHCII+ macrophages will continually recruit and prime cytotoxic CD4+ T cells, resulting in the disease pathology (Fig. 7).

## Discussion

The results of this report concur with previous studies showing that CD4+ T cells in the joint-footpad are mainly of the IFNγ-producing Th1-lineage and that their absence abrogates chikungunya severity (Carissimo et al, 2019; Teo et al, 2017; Teo et al, 2013). Removal of CD4+ T cells also reduced the presence of CD64+ macrophages and their precursors, the CD11b+Ly6C+ inflammatory monocytes (Carissimo et al, 2019). This is the first evidence, to our knowledge, that immune crosstalk exists between infiltrated CD4+ T cells and myeloid cells in CHIKV-infected joints, with IFNγ being one of the immune mediators involved in this interaction. IFNγ is an important cytokine involved in modulating the differentiation and activities of monocytes and macrophages (Castro et al, 2018; Delneste et al, 2003; Herbst et al, 2011; Luque-Martin et al, 2021). In this study, we detected high levels of IFNγ at the peak of CHIKV-induced joint-footpad inflammation, likely triggering differentiation of the infiltrated CD11b+Ly6C+ inflammatory monocytes into CD64+MHCII+ macrophages. This is supported

by our observations that (a) treatment of ex vivo peripheral blood cells with IFNγ stimulated CD11b+Ly6C+ monocytes to differentiate into a CD64+MHCII+ phenotype, and (b) in vivo IFNγ deficiency reduced this differentiation in the joint-footpad. Interestingly, IFNγ-secreting CD4+ T cells induced monocyte differentiation into inflammatory CD64+MHCII+ macrophages during leishmanial infection (De Trez et al, 2009; Romano et al, 2021). However, it is important to note that apart from IFNγ, other cytokines such as macrophage-colony stimulating factor (M-CSF) is also able to activate macrophages (Jones and Ricardo, 2013). Notably, our data would suggest a minor role for IFNγ during CHIKV infection, except for participating in the conversion of CD11b+Ly6C+ monocytes differentiation into CD64+MHCII+ macrophages. This lack of effect during IFNγ-deficiency may be due to a redundancy in the induction of type II interferon response genes (Wilson et al, 2017). Furthermore, CD4+ T cell numbers in IFNγ−/− animals were indifferent to the infected control animals (Fig. EV4F), may explain the similarity in disease progression between the two groups. Nevertheless, it is worthy to note that neutrophils were present in higher numbers in CHIKV-infected IFNγ−/− animals (Fig. EV4E). Neutrophils were previously reported to play a pathogenic role when the infiltration of monocytes/ macrophages into the joint-footpad was affected in chemokine (C-C motif) receptor 2 (CCR2) knockout animals (Poo et al, 2014). The role of neutrophils in CHIKV infection in the absence of IFNγ signaling remains to be elucidated.

Infiltrating CD4+ T cells in the joint-footpad were also capable of producing GM-CSF, a cytokine that has pleiotropic effects on myeloid cells and is often regarded as the central mediator of tissue inflammation (Becher et al, 2016; Croxford et al, 2015b; Zhan et al, 2019). However, only a subset of the IFNγ-producing CD4+CD44+ T cells was also producing GM-CSF. Interestingly this profile resembles that of the CD4+ T cells obtained from joints of rheumatoid arthritis

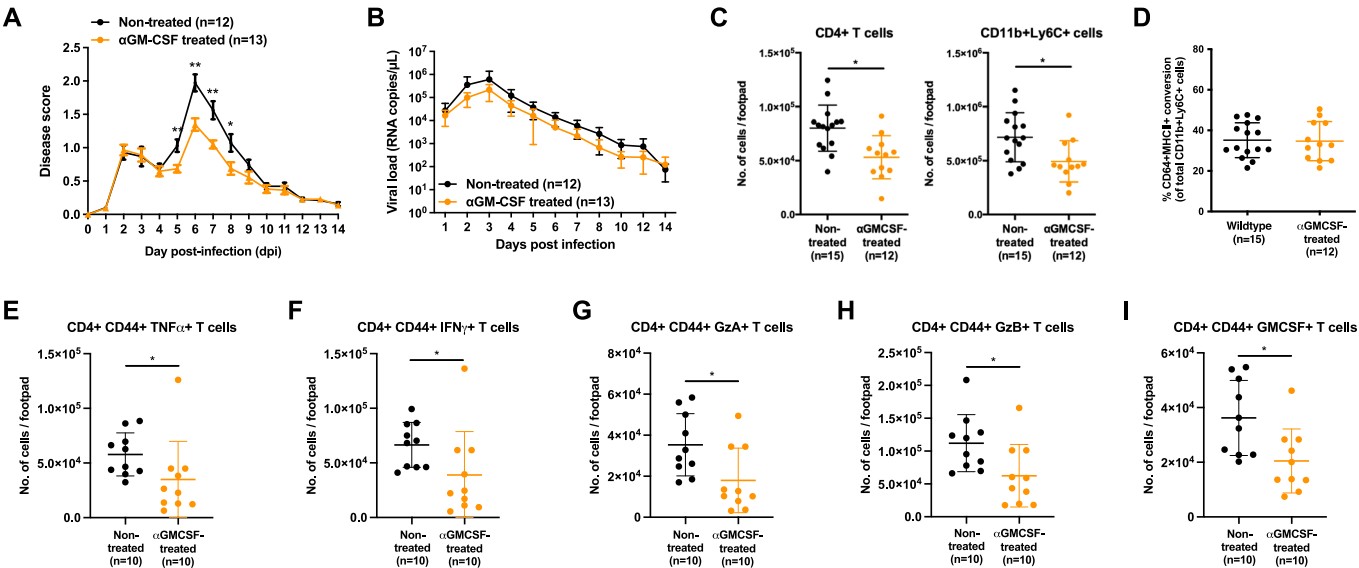

**Figure 3. depletion of GM-CSF reduces the severity of CHIKV infection.**

(A,B) Wild-type mice were infected with $1 \times 10^6$ PFU of CHIKV in the right footpad and subsequently treated with anti-GM-CSF antibodies at 4 dpi. Joint-footpad swelling (A) and viral RNA load (B) were monitored over 14 days. Data were from biological replicates obtained from two independent experiments. All data are presented as mean ± SD. Data comparisons were performed with non-parametric Mann–Whitney $U$ test (two-tailed). For footpad swelling: 5 dpi, **$P = 0.0064$; 6 dpi, **$P = 0.0011$; 7 dpi, **$P = 0.0068$; 8 dpi, *$P = 0.0372$. (C,D) Immunophenotyping of joint-footpad from anti-GM-CSF treated mice was performed at 6 days post-infection to determine the numbers of infiltrating CD4+ T cells and CD11b+Ly6C+ monocytes (C). Percentage differentiation of CD11b+Ly6C+ monocytes into CD64+MHCII+ macrophages in non-treated or GM-CSF-depleted CHIKV-infected animals is shown (D). Data were from biological replicates obtained from two independent experiments. All data are presented as mean ± SD. Data comparisons were performed with non-parametric Mann–Whitney $U$ test (two-tailed). CD4+ T cells, *$P = 0.037$; CD11b+Ly6C+, *$P = 0.0161$. (E–I) In a separate set of experiment, CHIKV infection was performed and treated with anti-GM-CSF antibodies at 4 dpi. At 6 dpi, numbers of TNFα- (E), IFNγ- (F), Granzyme A- (G), Granzyme B- (H), and GM-CSF- (I) producing CD4+CD44+ T cells were in the joint-footpad were quantified following stimulation with ionomycin and phorbol myristate acetate (PMA). Data were from biological replicates obtained from two independent experiments. All data are presented as mean ± SD. Data comparisons between the groups were performed with non-parametric Mann–Whitney $U$ test (two-tailed). TNFα-producing, *$P = 0.0175$; IFNγ-producing, *$P = 0.0138$, Granzyme A-producing, *$P = 0.0176$, Granzyme B-producing, *$P = 0.0175$, GMCSF-producing, *$P = 0.0138$. Source data are available online for this figure.

(RA) patients (Yamada et al, 2017). We also found that a subset of CD4+CD44+ T cells produced GM-CSF but not IFNγ (Figs. EV2B and C). Given that CD4+ T cells infiltrating the joint-footpad during CHIKV infection are primarily IFNγ-producing Th1 cells (Carissimo et al, 2019), the profile of these GM-CSF+, IFNγ- cells resembles that of ThGM (Zhang et al, 2013). At this juncture, it is important to note that GM-CSF can also be produced by cells such as macrophages, endothelial cells and fibroblasts (Shi et al, 2006). Nevertheless, differentiation into the CD64+MHCII+ phenotype does not seem to rely on GM-CSF during in vivo CHIKV infection. In fact, when GM-CSF was depleted, the percentage of differentiated CD64+MHCII+ macrophages present was comparable to the control animals (Fig. 3D). Instead, the results of this study suggest that GM-CSF is more likely to be involved in recruiting both CD4+ T cells and CD11b+Ly6C+ precursor inflammatory cells into the joint-footpad, as their numbers were significantly reduced following GM-CSF neutralization (Fig. 3C).

Consistent with this, GM-CSF has been shown to participate in the recruitment of myeloid cells into inflammatory foci during neuroinflammation (Croxford et al, 2015a) and to mediate recruitment of both CD4+ and CD8+ T cells into the lungs of ovalbumin-exposed mice (Stampfli et al, 1998). It was also reported that GM-CSF boosts cytokine production of both Th1 and Th2 CD4+ T cells (Zhang et al, 2013). Concordantly, our treatment of CHIKV-infected mice with anti-GM-CSF antibodies significantly reduced joint-footpad swelling at 6 dpi, demonstrating a

therapeutic effect. Depletion of GM-CSF also resulted in a global reduction in the numbers of inflammatory CD4+ T cells producing TNFα, IFNγ, Granzyme A and Granzyme B. Granzyme A and B are two serine proteases which could participate in CHIKV-associated tissue inflammation and edema (Schanoski et al, 2019; Teo et al, 2015; Wilson et al, 2017). GM-CSF may also upregulate CCL17 expression in macrophages, thereby mediating inflammatory arthritis and associated inflammatory pain (Achuthan et al, 2016) and CCL17 levels were also significantly reduced following anti-GM-CSF therapy in RA patients (Guo et al, 2019). GM-CSF has also been reported to enhance expression of IL-6, IL-1, and TNF in pro-inflammatory macrophages and microglia (Na et al, 2016; Parajuli et al, 2012). These three pro-inflammatory cytokines were also detected at high levels in the CHIKV-infected joint-footpad. Given the importance of GM-CSF during CHIKV infection, its potential as a future CHIKV therapeutic target is further justified by clinical trials evaluating GM-CSF blockade in RA patients (Burmester et al, 2017; Taylor et al, 2019).

Upregulation of MHCII expression by CD64+ macrophages suggests a reciprocal interaction with the CD4+ T cells in the footpad, leading to their activation and production of more IFNγ and GM-CSF. This assumption is supported by our transcriptomics data comparing CD64+MHCII- and CD64+MHCII+ macrophages, with GO analysis highlighting that the "T-helper cell differentiation" and "antigen processing and presentation" functional groups

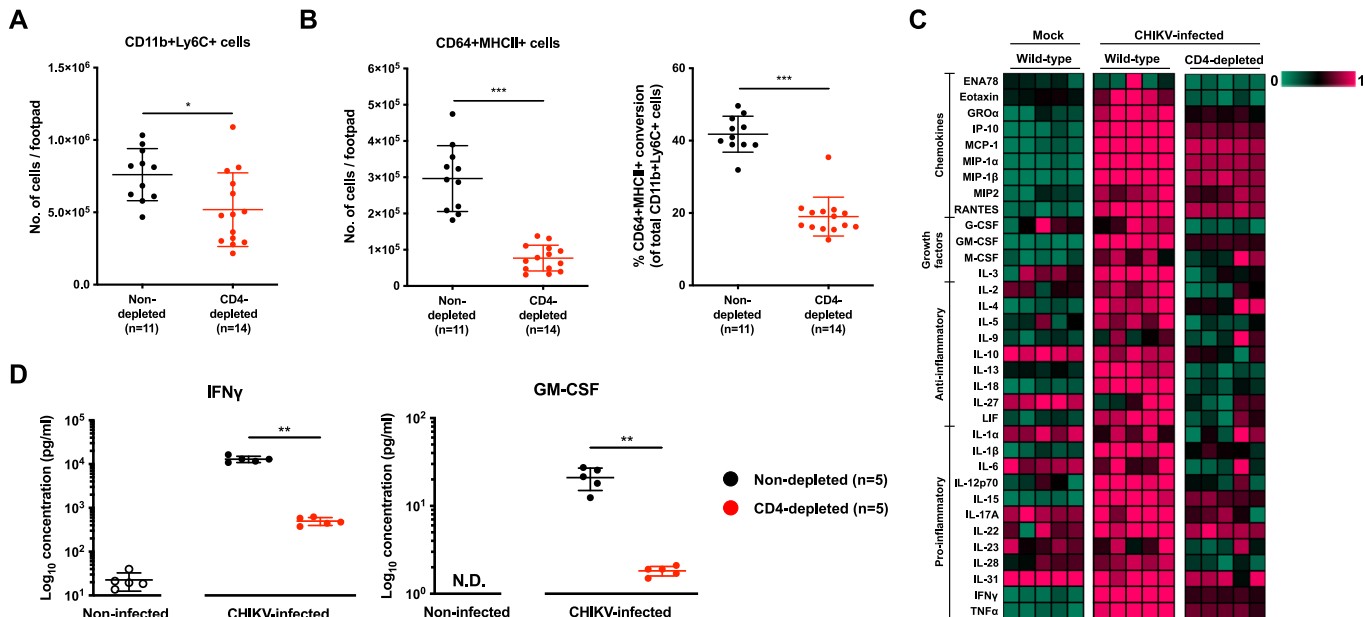

**Figure 4. CD4+ T cells depletion alters levels of critical immune mediators in the CHIKV-infected joint-footpad.**

(A,B) Wild-type animals were infected with 1 × 10⁶ PFU of CHIKV in the right footpad, and anti-CD4 antibodies were given intraperitoneally at -1 and 4 days post-infection (dpi). Immunophenotyping was performed at 6 dpi to determine the numbers of infiltrating CD11b⁺Ly6C⁺ monocytes in CD4-depleted joint-footpads (A). Percentage differentiation of CD11b⁺Ly6C⁺ monocytes into CD64⁺MHCII⁻ and CD64⁺MHCII⁺ macrophages and absolute counts of CD64⁺MHCII⁺ macrophages in CHIKV-infected non-CD4-depleted or CD4-depleted animals (B). Data presented were from biological replicates obtained from two independent experiments and presented as mean ± SD. Data comparisons between the groups were performed with non-parametric Mann–Whitney $U$ test (two-tailed). For CD1b⁺Ly6C⁺ (A), *$P = 0.0152$; For CD64⁺MHCII⁺ (B), *$P = 0.000000449$; for CD64⁺MHCII⁺ conversion (B), ***$P = 0.000000897$. (C,D) Joint lysates were obtained from CHIKV-infected CD4-depleted, infected wild-type (non-CD4-depleted) mice, and mock-infected control mice at peak chikungunya joint pathology (6 dpi). A multiplex microbead-based assay was used to quantify the levels of immune mediators present in these samples. Heatmap showing the levels of the analyzed immune mediators in each group of animals (C). Dot plots showing the absolute quantities of IFNγ and GM-CSF, highlighting the significant differences between the various groups of animals (D). Data were from biological replicates and presented as mean ± SD. Data comparisons between the groups were performed with non-parametric Mann–Whitney $U$ test (two-tailed). IFNγ, **$P = 0.0079$; GMCSF, **$P = 0.0079$. Source data are available online for this figure.

contained 18 of the 89 identified DEGs between these two cell types. Moreover, deletion of MHCII is known to reduce joint-footpad swelling during CHIKV infection (Nakaya et al, 2012).

While we are limited by the lack of research tools in effectively depleting away the CD64⁺MHCII⁺ macrophages from CHIKV-infected animals, clodronate-liposome treatment at 5 dpi significantly reduced joint-footpad swelling, coupled with a reduced presence of both CD64⁺ macrophages and CD4⁺ T cells (Fig. 6C,D). The reduced CD4⁺ T cell numbers could be attributed to their reduced induction and expansion, following the depletion of macrophages, rather than a decreased T cell infiltration given that the levels of IP-10 (Fig. 6E) were not affected by clodronate-liposome treatment.

The use of clodronate-liposome is a common tool used to investigate in vivo functions of macrophages (Nguyen et al, 2021). Here, clodronate was specifically given at 5 dpi to minimize any impact from phagocytic cell absence during innate and early adaptive immune responses. Depletion just before peak of CHIKV disease, when CD64⁺ macrophage infiltration and differentiation is high, demonstrated indirectly the importance of in vivo CD64⁺MHCII⁺ macrophages in mediating the priming CD4⁺ T cells during CHIKV immunopathogenesis. This particular approach, due to the lack of better tools, was aimed to indirectly correlate the importance of macrophages in mediating CHIKV pathology along with the CD4⁺ T cells. The presence of such a concerted effort between MHCII⁺ macrophages and CD4⁺ T cells has been reported in adipose tissue meta-inflammation (Cho et al, 2014).

Another limitation in our study was that an isotype antibody was not utilized as a control in our antibody-based in vivo depletion experiments as we wanted the infection to resemble closely to that of the non-treated animals. Nevertheless, we have shown in Appendix Fig. S1, disease severity and viral RNA load clearance were comparable between animals receiving PBS and isotype control antibodies.

Among the identified DEGs, five particularly important genes—CCR7, IL12B, CD74, TBX21, and SERPINB9—were more abundant in CD64⁺MHCII⁺ than in CD64⁺MHCII⁻ macrophages. CD74 was shown to mediate alphaviral arthritis (Herrero et al, 2013) and neutrophil recruitment during inflammation (Takahashi et al, 2009). CCR7/CCL21 signaling in synovial macrophages was reported to mediate joint inflammation and osteoclast formation in RA (Van Raemdonck et al, 2020). TBX21 (also called T-bet) is a well-characterized Th1-specific transcription factor (Szabo et al, 2000). Its expression in macrophages (Lighvani et al, 2001), is thought to participate in regulating IL-6 expression in bone marrow-derived macrophages (Hayashi et al, 2019). IL-12B (also known as IL-12p40) is a subunit of the cytokine IL-12 (Kobayashi et al, 1989), which is involved in polarizing naive CD4⁺ T cells towards a Th1 phenotype and inducing IFNγ production in T and NK cells during inflammation (Trinchieri, 1995). IL-12 is associated with increased disease severity in CHIKV patients (Chow et al, 2011; Kelvin et al, 2011; Teng et al, 2015). Finally, Serpinb9 is an active inhibitor of granzyme B (Bird et al, 1998) and

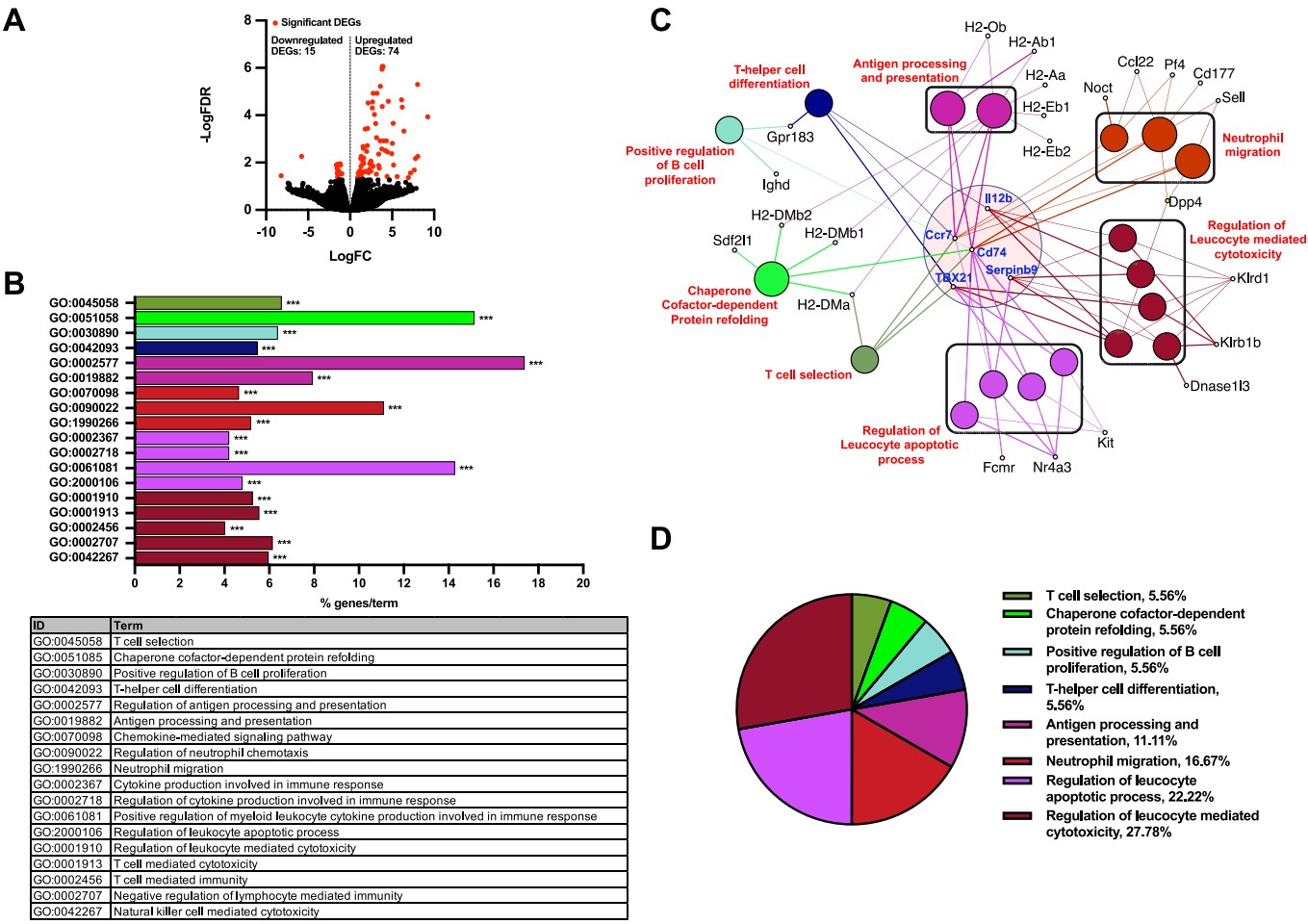

Figure 5. RNA-seq analyses of CD64⁺MHCII⁺ and CD64⁺MHCII⁻ macrophages reveal differentially expressed genes that influence multiple immunological aspects.

(A) CD64⁺MHCII⁺ and CD64⁺MHCII⁻ macrophages were sorted from CHIKV-infected joints ($n = 5$ animals per group) at 6 dpi and processed for RNA-seq. A total of 89 DEGs were identified. (B) Representative gene ontology terms and pathways are depicted, including genes involved in Th1 and Th2 differentiation as well as antigen processing and presentation. Analysis was carried out with a two-sided hypergeometric test, corrected with Bonferroni stepdown. ***$P = <$ 0.001 (refer to Dataset EV1 for exact $P$ values). (C) Gene ontology terms are presented as nodes and clustered together based on the similarity of genes present in each term or pathway. Ccr7, Cd74, Il12b, TBX21, and Serpinb9 are identified as genes highly associated with many of these terms. (D) A functionally grouped network of enriched pathways or terms based on the 89 DEGs were generated by querying the Gene Ontology - biological database with ClueGo. Source data are available online for this figure.

it's expression in CD64⁺ macrophages, potentially protects themselves against the cytotoxic effects of granzyme B present in inflamed tissue (Teo et al, 2015).

In humans, CD4⁺ T cells are detected alongside the CD8⁺ T cells in the synovial and muscle biopsies of patients in the chronic phase of the disease (Hoarau et al, 2010; Ozden et al, 2007). Particularly, the CD4⁺ T cells were postulated to induce inflammation through the production of pro-inflammatory cytokines (Hoarau et al, 2010; Petitdemange et al, 2015). However, the exact roles of human CD4⁺ T cells in CHIKV immunopathogenesis remains under-explored (Mapalagamage et al, 2022). On the other hand, macrophages were similarly identified in synovial and muscle biopsies of patients (Hoarau et al, 2010; Petitdemange et al, 2015). In fact, macrophages and monocytes are known target cells of CHIKV (Her et al, 2010) and has been suggested to be a reservoir for chronic infection in humans (Fox and Diamond, 2016). Studies using relevant human monocytic cell lines (Felipe et al, 2020; Guerrero-Arguero et al,

2020; Srivastava et al, 2023), as well as with primary human monocytes-derived macrophages (Lau et al, 2023), reported a pro-inflammatory immune response following CHIKV infection.

In conclusion, this study describes a pathogenic immune crosstalk between CD64⁺MHCII⁺ macrophages and CD4⁺ T cells in chikungunya disease progression and offers new directions for the future development of anti-CHIKV drugs targeting the CD64⁺MHCII⁺ macrophages as an alternative therapeutic targets.

# Methods

## Ethics approval

Three- to four-week-old gender-matched wild-type or IFN-gamma (IFNγ) deficient mice in C57BL/6J background were used in all in vivo experiments. Animals at this age exhibit pronounced

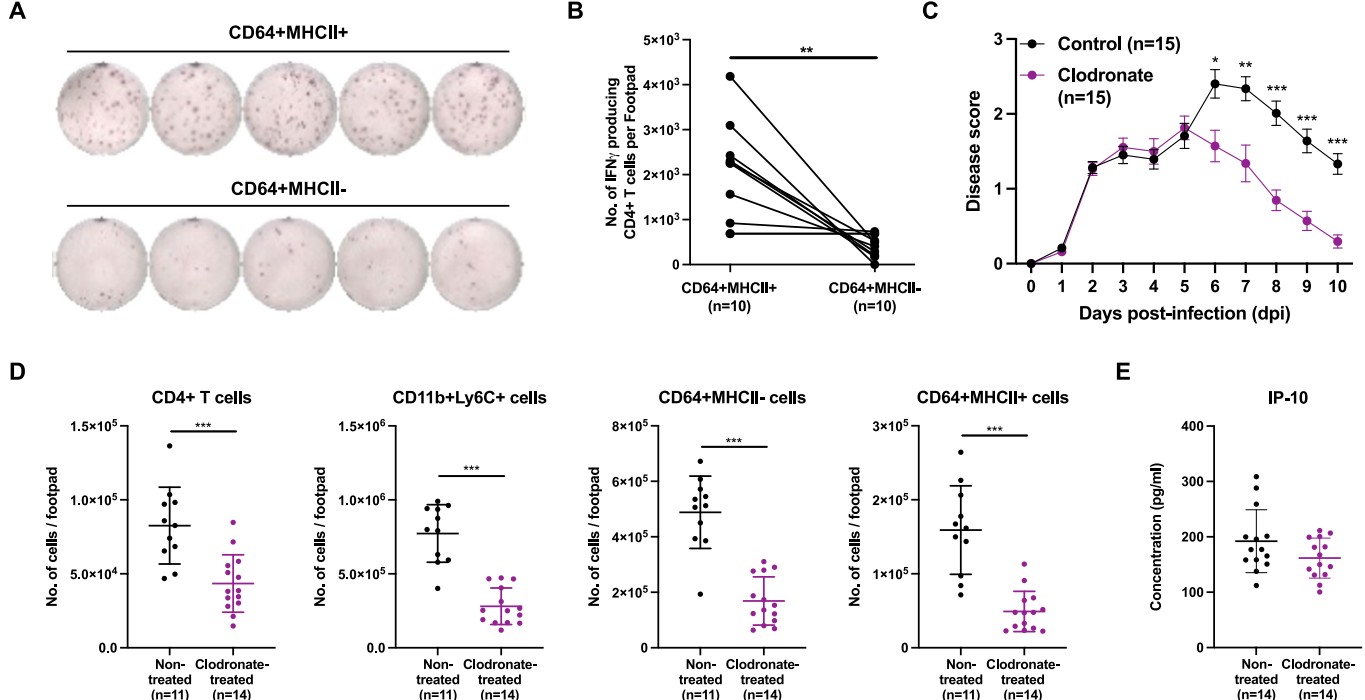

**Figure 6. Crosstalk between CD64⁺MHCII⁺ macrophages and CD4⁺ T cells drive CHIKV pathogenesis.**

(A,B) CD64⁺MHCII⁺ and CD64⁺MHCII⁻ macrophages were sorted and CHIKV-specific CD4⁺ T cells were isolated from joint-footpad of CHIKV-infected animals at 6 dpi. CHIKV-specific CD4⁺ T cells were subsequently restimulated with CHIKV antigen in the presence of either CD64⁺MHCII⁺ or CD64⁺MHCII⁻ macrophages for 18 h. Representative ELISpot images from 5 animals per group are shown (A). Paired line graph showing the numbers of IFNγ-producing CHIKV-specific CD4⁺ T cells post-re-stimulation (B). Data were from biological replicates obtained from two independent experiments and comparison was performed with non-parametric Wilcoxon matched-pairs signed rank test (two-tailed; **$P = 0.0078$). (C) Wild-type animals were infected with $1 \times 10^6$ PFU of CHIKV in the right footpad. At 5 dpi, animals were given intraperitoneally with either clodronate liposome (1 mg) or empty liposome (control). Joint-footpad swelling of these animals were monitored over a period of 10 days. Data were from biological replicates obtained from two independent experiments and presented as mean ± SD. Data comparison between groups was performed with non-parametric Mann–Whitney $U$ test (two-tailed). 6 dpi, *$P = 0.0102$; 7 dpi, **$P = 0.0051$; 8 dpi, ***$P = 0.0000551$; 9 dpi, ***$P = 0.0000535$; 10 dpi, ***$P = 0.00000135$. (D,E) Wild-type animals were infected with $1 \times 10^6$ PFU of CHIKV in the right footpad. At 5 dpi, animals were given intraperitoneally with either clodronate liposome (1 mg) or empty liposome (control). Immunophenotyping was performed at 6 dpi for both groups of animals. Graphs show the numbers of CD4⁺ T cells, CD11b⁺Ly6C⁺ precursor cells, CD64⁺MHCII⁻ and CD64⁺MHCII⁺ macrophages present in the joint-footpad (D). Levels of IP-10 present in joint-footpad was quantified at 6 dpi from another set of animals (E). Data were from biological replicates obtained from two independent experiments and presented as mean ± SD. Data comparisons between the groups were performed with non-parametric Mann–Whitney $U$ test (two-tailed). CD4⁺ T, ***$P = 0.0005$; CD11b⁺Ly6C⁺, ***$P = 0.00000538$; CD64⁺MHCII⁻, ***$P = 0.00000538$; CD64⁺MHCII⁺, ***$P = 0.00000853$. Source data are available online for this figure.

joint-footpad swelling after CHIKV infection. All experimental procedures were approved by the Institutional Animal Care and Use Committee (IACUC; #211635) of the Agency for Science, Technology and Research (A*STAR), Singapore, in accordance with the guidelines of the Agri-Food and Veterinary Authority and the National Advisory Committee for Laboratory Animal Research of Singapore. Animals were purchased from Jackson Laboratory and were further bred and housed under specific pathogens-free conditions in the Biological Resource Center of A*STAR.

## Virus stock

CHIKV-SGP11 was isolated from an infected patient during an outbreak in Singapore in 2008 (Her et al, 2010) and was propagated in C6/36 cells (CRL-1660, ATCC©). C6/36 cells were cultured in Leibovitz's L-15 medium (ThermoFisher Scientific) containing 10% fetal bovine serum (FBS; Cytiva). Propagated CHIKV was subsequently purified by ultra-centrifugation (Kam et al, 2012).

Titration of the purified virus was subsequently performed with VeroE6 (CRL-1586, ATCC©) before being used in infection studies (Her et al, 2010; Kam et al, 2012). VeroE6 cells were cultured in Dulbecco's modified Eagle medium (DMEM; Gibco) containing 10% FBS (Cytiva). Purified CHIKV virus was aliquoted and kept in $-80\,°C$ for long-term storage. Both cell lines were tested for mycoplasma.

## CHIKV infection

Mice were inoculated subcutaneously with $1 \times 10^6$ plaque-forming units (PFU) of CHIKV isolate (diluted in 30 μL of PBS) in the ventral side of the right hind footpad towards the ankle. Viral RNA load and joint-footpad swelling of the infected animals were monitored as previously described (Carissimo et al, 2019; Her et al, 2015; Kam et al, 2012; Lee et al, 2015; Lum et al, 2013; Teng et al, 2012; Teo et al, 2015; Teo et al, 2013). Viral RNA load was determined from 10 μL of tail blood diluted in 130 μL of PBS-citrate solution. Joint-footpad swelling is presented as disease score

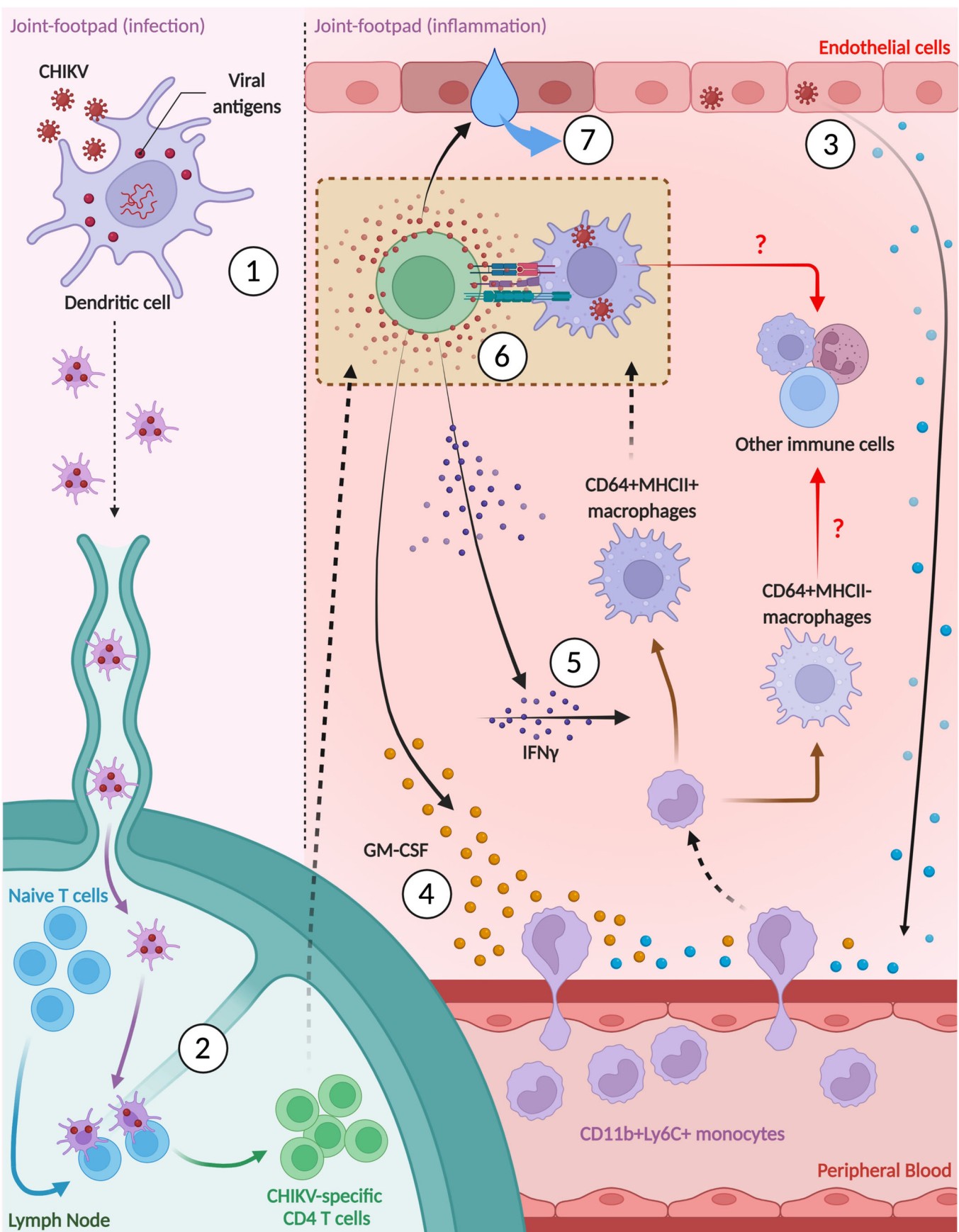

◀ **Figure 7.   Crosstalk between CD64⁺MHCII⁺ macrophages and CD4⁺ T cells drive CHIKV pathogenesis.**

Following active CHIKV infection, (1) viral antigens are first up taken by residential dendritic cells which then travel to the draining lymph nodes. (2) Here, CD4⁺ T cells are being primed and activated. (3) Initial release of chemokines by CHIKV-infected non-immune cells (e.g., endothelial and fibroblastic cells) subsequently lead to (4) recruitment of CD4⁺ T cells and CD11b⁺Ly6C⁺ inflammatory monocytes along chemokine gradients. Recruited CD4⁺ T cells further participate in recruitment of CD11b⁺Ly6C⁺ inflammatory monocytes through secretion of GM-CSF. In addition, (5) IFNγ secreted by CD4⁺ T cells aid in the activation and differentiation of infiltrated CD11b⁺Ly6C⁺ inflammatory monocytes into CD64⁺MHCII⁺ macrophages. (6) CD64⁺MHCII⁺ macrophages reciprocally interact and activate with CD4⁺ T cells by acting as antigen-presenting cells. (7) Eventually, activated CD4⁺ T cells cause CHIKV-induced joint-footpad pathology (e.g., edema) through secretion of numerous pathogenic mediators (e.g., TNFα, Granzyme-A or Granzyme-B). Source data are available online for this figure.

reflecting the changes in footpad size post-CHIKV infection relative to pre-infection.

## CHIKV viral RNA determination

Viral RNA was obtained from 140 μL of PBS-citrate-diluted blood using QIAamp Viral RNA Kit (QIAGEN). Viral RNA load quantification was performed by qRT-PCR using QuantiTect Probe RT-PCR Kit (QIAGEN) modified from a previously described method to detect negative-strand nsP1 RNA (Plaskon et al, 2009) Sequences of primers and probes are: Forward primer (5′-GGCAG TATCGTGAATTCGATGCGACACGGAGACGCCAACATT-3′), reverse primer (5′-AATAAATCATAAGTCTGCTCTCTGTCTAC ATGA-3′) and probe (5′-[6FAM]TGCTTACACACAGACGT[T AM]-3′). Reactions were performed in 12.5 μL reaction volume using Applied Biosystems (ABI) 7900HT Fast Real-Time PCR System, with the following conditions: (1) reverse transcription step (50 °C for 30 min; 1 cycle); (2) PCR initial activation step (95 °C for 15 min; 1 cycle); (3) 2-step cycling (94 °C for 15 s, follow by 60 °C for 1 min; 45 cycles). CHIKV viral RNA load was estimated from a standard curve generated using serial dilution of synthetic CHIKV negative-strand nsP1 RNA transcripts as described previously (Teng et al, 2012).

## Leukocyte profiling

Joint-footpads were excised 6 days post-infection (dpi) and were digested in RPMI (Hyclone) with 10% heat-inactivated Fetal Bovine Serum (FBS; Biowest), supplemented with DNase I (50 mg/mL; Roche Applied Science), collagenase IV (20 mg/mL; Sigma-Aldrich), and dispase (2 U/mL; Gibco). The digestion was performed with slight agitation at 37 °C for 3 h before being filtered through a cell strainer (40 μm pore size). Red blood cells were subsequently lysed with Flow Cytometry Mouse Lyse Buffer (R&D) and cells were purified with Percoll solution (Sigma-Aldrich) diluted (35% vol/vol) in RPMI FBS. Isolated cells were enumerated and stained with the LIVE/DEAD determination dye (ThermoFisher Scientific) for 20 min. Stained cells were washed and blocked with staining media (PBS containing 1% (vol/vol) rat and mouse serum) for 5 min before being stained with the following antibodies: Rat anti-mouse CD45 (clone: 30-F11; dilution 1:400), Rat anti-mouse CD3 (clone: 17A2; dilution 1:200), Rat anti-mouse CD4 (clone: GK1.5; dilution 1:400), Rat anti-mouse CD8 (clone: 53.6-7; dilution 1:200), Rat anti-mouse B220 (clone: RA3-6B2; dilution 1:200), Mouse anti-mouse NK1.1 (clone PK136; dilution 1:400), Rat anti-mouse CD11b (Clone: M1/70; dilution 1:400), Hamster anti-mouse CD11c (clone: N418; dilution 1:400), Rat anti-mouse Ly6C (clone: HK1.4; dilution 1:400), Rat anti-mouse Ly6G (clone: 1A8; dilution 1:400), Mouse anti-mouse CD64 (clone: X54-4/ 7.1; dilution 1:200), Rat anti-mouse MHCII (clone: M5/114.15.2;

dilution 1:400) and Rat anti-mouse LFA-1 (clone: H155-78; dilution 1:200). For NK1.1 staining, a counterstain was performed with BUV737-conjugated streptavidin (dilution 1:400). All antibodies used for staining were purchased from BD Biosciences, BioLegend or ThermoFisher Scientific. Stained cells were fixed with eBioscience IC Fixation Buffer (ThermoFisher Scientific) and acquired on a LSRII Flow Cytometer (BD Biosciences). Analyses were performed with FlowJo (Tree Star) version 10.1.

## T-cell stimulation assay

Footpad cells were harvested as mentioned above. Subsequently, an aliquot of the isolated cells were incubated in 50 μL of IMDM (Hyclone) with 10% FBS (Hyclone) and 1% Penicillin and Streptomycin (Sigma-Aldrich) with or without 40 ng/mL PMA (Sigma-Aldrich) and 1 μg/mL Ionomycin (Sigma-Aldrich) at 37 °C. After 1 h, 1X Brefeldin A (BioLegend) and 1X GolgiStop (BD Biosciences) were added and further incubated for another 3 h at 37 °C. Stimulated cells were collected, washed, and stained with the LIVE/DEAD determination dye (Thermo Fisher Scientific) for 10 min. Cells were washed and blocked with staining media for 5 min before being stained with the following antibodies targeting surface markers: Rat anti-mouse CD45 (clone: 30-F11; dilution 1:400), Mouse anti-mouse NK1.1 (clone: PK136; dilution 1:400), Rat anti-mouse CD19 (clone: 6D5; dilution 1:400), Rat anti-mouse Ter119 (clone: Ter119; dilution 1:400), Rat anti-mouse CD3 (clone: 17A2; dilution 1:200), Rat anti-mouse CD4 (clone: RM4-5; dilution 1:200) and Rat anti-mouse CD44 (clone: IM7; dilution 1:400). For biotinylated NK1.1, CD19 and Ter119, a secondary stain was performed with BUV737-conjugated streptavidin (dilution 1:400). All antibodies used for staining were purchased from BD Biosciences, BioLegend or ThermoFisher Scientific. Staining was performed for 30 min on ice. Cells were washed, fixed, and permeabilized with eBioscience FOXP3/Transcription Factor Staining Buffer Set (ThermoFisher Scientific) following the manufacturer's protocol. Intracellular staining was subsequently performed for 30 min on ice with the following antibodies diluted in the Perm/Wash buffer containing 1% (vol/vol) rat and 1% mouse serum: Rat anti-mouse IFNγ (clone: XMG1.2; dilution 1:400), Rat anti-GM-CSF (clone: MP122E9; dilution 1:400), Rat anti-TNFα (clone: MP6-XT22; dilution 1:400), Rat anti-Granzyme A (clone: 3G8.5; dilution 1:400) and Rat anti-Granzyme B (clone: QA16A02; dilution 1:400) antibodies. All antibodies used for staining were purchased from BD Biosciences, BioLegend, or ThermoFisher Scientific. Stained cells were then washed and immediately acquired on a LSRII Flow Cytometer (BD Biosciences). Analyses were performed with FlowJo (Tree Star) version 10.1. Data were back-calculated and expressed as the number of positive cells per footpad.

## Peripheral blood-cell stimulation assay

Aliquots of naive mouse whole blood were obtained from the tail. Briefly, 10 μL of blood was mixed with an equal volume of PBS-citrate to prevent clotting. Following, red blood cells were lysed and the remaining cells were subsequently incubated in 50 μL of IMDM (Hyclone) with 10% FBS (Hyclone) and 1% Penicillin and Streptomycin (Sigma-Aldrich). Cells were then stimulated with either IFNγ (STEMCELL Technologies) or GM-CSF (STEMCELL Technologies), individually or together, at a fixed concentration of 25 ng/mL. Stimulation was allowed over a period of 24 h. Stimulated cells were subsequently washed and stained with the LIVE/DEAD determination dye (Thermo Fisher Scientific) for 20 min. Stained cells were washed and blocked with staining media (PBS containing 1% (vol/vol) rat and mouse serum) for 5 min before being stained with the following antibodies: Rat anti-mouse CD45 (clone: 30-F11; dilution 1:400), Rat anti-mouse CD11b (Clone: M1/70; dilution 1:400), Rat anti-mouse Ly6C (clone: HK1.4; dilution 1:400), Rat anti-mouse Ly6G (clone: 1A8; dilution 1:400), Mouse anti-mouse CD64 (clone: X54-4/7.1; dilution 1:200) and Rat anti-mouse MHCII (clone: M5/114.15.2; dilution 1:400). All antibodies used for staining were purchased from BD Biosciences, BioLegend or ThermoFisher Scientific. Stained cells were fixed with eBioscience IC Fixation Buffer (ThermoFisher Scientific) and acquired on a LSRII Flow Cytometer (BD Biosciences). An additional staining was performed on freshly obtained naive mouse peripheral blood cells to demarcate the positive gatings for MHCII and CD64 expression on gated $CD11b^+Ly6C^+$ cells. Analyses were performed with FlowJo (Tree Star) version 10.1.

## Multiplex immunoassay

At 6 dpi, mice were sacrificed by terminal anesthesia with ketamine [150 mg/kg]/xylazine [10 mg/kg] followed by intra-cardial perfusion with PBS. Joint-footpads were then collected and homogenized in 1 mL RIPA buffer (50 mM Tris-HCl, pH 7.4; 1% NP-40; 0.25% sodium deoxycholate; 150 mM NaCl; and 1 mM EDTA) with 1X protease inhibitor (Roche Holding AG) using the gentleMACS M Tube and gentleMACS Dissociator (Miltenyi Biotec). Cell lysates were sonicated at 70% intensity (Branson Ultrasonics Sonifier S-450) for 15 s on ice before being spun down at 12,000 rpm. Supernatants were eventually collected for downstream immune mediators quantification with the Procarta Mouse Cytokine and Chemokine 36-plex immunoassay (Thermo Fisher Scientific). Preparations of plasma samples and reagents, as well as immunoassay procedures were performed according to the manufacturer's protocol. The cytokines and chemokines assayed included IFNγ, IL-12p70, IL-13, IL-1β, IL-2, IL-4, IL-5, IL-6, TNFα, GM-CSF, IL-18, IL-10, IL-17A, IL-22, IL-23, IL-27, IL-9, GROα, IP-10, MCP-1, MCP-3, MIP-1α, MIP-1β, MIP2, RANTES, Eotaxin, IFNα, IL-15, IL-28, IL-31, IL-1α, IL-3, G-CSF, LIF, ENA-78/CXCL5, and M-CSF. IL-3 and IFNα were not detected and were thus excluded from any analyses. Data were acquired using a Luminex FlexMap 3D instrument (Luminex) and analyzed using Bio-Plex Manager 6.0 software (Bio-Rad) based on standard curves plotted through a five-parameter logistic curve setting. Quantities of detected immune mediators are presented with a heat map done using TM4-MeV and plotted using $log_{10}$ concentrations values scaled between 0 and 1 for visualization.

## Drug treatment

For $CD4^+$ T cells depletion, mice were inoculated intraperitoneally with 500 μg of anti-mouse CD4 antibody (Clone: GK1.5; Bio X Cell) at −1 and 4 dpi. For GM-CSF depletion, mice were inoculated intraperitoneally with 250 μg of anti-mouse GM-CSF antibodies (Clone: MP1-22E9; Bio X Cell) at 4 dpi. For IFNγ depletion, mice were inoculated intraperitoneally with 500 μg of anti-mouse IFNγ antibody (Clone: XMG1.2; Bio X Cell) at 0, 2, and 4 dpi. All drugs given were prepared in 100 μL volume diluted in sterile PBS. Control animals were given 100 μL of PBS, to ensure that in these animals, the infection will resemble as closely as possible to non-treated animals. Appendix Fig. S1 shows that disease severity and viral load clearance were comparable between animals receiving PBS and isotype control antibodies. Macrophages were depleted with clodronate liposomes (LIPOSOMA) given intravenously at 5 dpi. Clodronate liposomes were given at a final concentration of 1 mg/mL, in a volume of 200 μL. Control animals were given empty liposomes (LIPOSOMA).

## CyTOF profiling

CHIKV-infected mouse joint-footpads were harvested at 6 dpi and processed as described above. Myeloid cells were then enriched by the removal of T cells with CD90.2 microbeads (Miltenyi Biotec) following the manufacturer's protocol. Recovered cells (~3 million cells) were first stained with 200 μM Cisplatin (Sigma-Aldrich) for 5 min to exclude dead cells. Cells were subsequently washed and stained in 50 μL volume of antibody cocktail targeting specific myeloid surface markers (Becher et al, 2014). Staining was done on ice for 30 min before being washed. Stained cells were then fixed in freshly prepared 2% paraformaldehyde (PFA) before being barcoded. Barcoding was performed as previously described (Becher et al, 2014). Samples were eventually washed and resuspended in MilliQ water for acquisition on the CyTOF machine (Newell et al, 2012). Data were exported in flow-cytometry file format and barcode identities were deconvoluted using FlowJo software (Tree Star) (version 10.1). Pre-gated live $CD45^+$ cells were randomly down-sampled to 5000 events for each sample. Logical transformation was then performed using autoLgcl of the Bioconductor package cytofkit2. The dimensionality algorithm Uniform Manifold Approximation and Projection for Dimension Reduction (UMAP) (Becht et al, 2019) was then run using 1000 max iterations, 30 nearest neighbors, and other default parameters (Becht et al, 2019; Chen et al, 2016). PhenoGraph (R package Rphenograph version 0.99.1) was used with default parameters (k = 30) for clustering.

## Sorting of targeted cells

Joint-footpad cells at 6 dpi were harvested as described above. After the digestion and purification steps, joint-footpad cells were stained the LIVE/DEAD determination dye (Thermo Fisher Scientific) for 20 min. Thereafter, the cells were stained with the following antibodies: Rat anti-mouse CD45 (clone: 30-F11; dilution 1:400), Rat anti-mouse CD11b (Clone: M1/70; dilution 1:400), Rat anti-mouse Ly6C (clone: HK1.4; dilution 1:400), Rat anti-mouse Ly6G (clone: 1A8; dilution 1:400), Mouse anti-mouse CD64

(clone: X54-4/7.1; dilution 1:200) and Rat anti-mouse MHCII (clone: M5/114.15.2; dilution 1:400). All antibodies used for staining were purchased from BD Biosciences, BioLegend or ThermoFisher Scientific. Stained cells were washed, resuspended in an appropriate buffer and the target cells of CD64$^+$MHCII$^-$ and CD64$^+$MHCII$^+$ phenotype were sorted with FACS Aria III sorter (BD Biosciences). Sorted cells were spun down and used for ELISpot. Alternatively, the sorted cells were resuspended in Trizol (ThermoFisher Scientific) for downstream total RNA extraction.

## Radiochemistry of [$^{18}$F]FEPPA and PET imaging radiochemistry

The [$^{18}$F]FEPPA was prepared using a one-pot reaction as described previously (Hartimath et al, 2019). In brief, cyclotron produced no-carrier-aqueous [$^{18}$F] fluoride was trapped on a preconditioned Sep-Pak light QMA cartridge (Waters). The trapped [$^{18}$F] fluoride was eluted using a mixture 96: 4 (v/v) acetonitrile: water mixture containing $K_2CO_3$ and Kryptofix 222. The Kryptofix complex was dried azeotropically and the dried [$^{18}$F] fluoride was reacted with FEPPA precursor (5 mg, 9.4 mmol, 2-(2-((N-(4-phenoxypyridin-3-yl) acetamido) methyl)phenoxy) ethyl-4-methylbenzenesulfonate) in an anhydrous acetonitrile (0.7 mL) and the reaction was further heated at 95 °C for 10 min under nitrogen condition. The reaction mixture was cooled and crude mixture was diluted with 4 mL of HPLC mobile phase (30/70 v/v ethanol/0.1% phosphoric acid in water) and purified using a isocratic semi-preparative radio-HPLC (Nucleodur pyramid C18, 5 m, 110 Å, 250 × 10 mm; 5 mL/min, λ = 254 nm). At 13.4 to 13.6 min, a single peak of [$^{18}$F] FEPPA was collected, and the fraction was diluted to 10 mL with 0.9% w/v saline and neutralized with 10% sodium bicarbonate. The mixture was filtered through a 0.22 mm Millex GV filter before injecting into the animals. Analytical radio-HPLC was carried out using a UFLC Shimadzu HPLC system fitted with a dual wavelength UV detector and a NaI/PMT-radio detector (Flow-Ram, LabLogic). A CRC-55tPET dose calibrator was used to measure radioactivity (Capintec, USA). At the end of the synthesis, the radiochemical purity was greater than 99%, and the specific activity was 110 ± 23 GBq/mmol ($n = 4$).

## Small animal PET/CT imaging

PET/CT scanning was performed using Siemens Inveon PET/CT scanner on days 0, 3, and 6 following CHKV infection. In brief, animals were anesthetized with 1.5 percent inhaled isoflurane before injected with [$^{18}$F]FEPPA (~15MBq each animal) via the lateral tail vein. After 60 min of post-injection, a static PET scan of 10 min was performed followed by a 5 min single frame CT scan was acquired for anatomical co-registration and attenuation correction. During the scanning, a Biovet physiological monitoring system was used to track the animal body temperature and breathing rate. After attenuation and decay correction, list mode PET data was reconstructed using the 3D ordered subset expectation maximization (OSEM 3D) algorithm with 16 subsets and 2 iterations. For anatomical delineation, the calibrated PET images were co-registered with CT images, and quantitative radioactivity in the area of interest was determined by inserting a ROI using Amide software (version 10.3 Sourceforge). The [$^{18}$F]FEPPA uptake was measured in the paw area and represented as a percentage of the administered dose per gram of tissue in the ROI (% ID/g).

## RNA-sequencing

Total RNA was extracted using the Arcturus PicoPure RNA Isolation kit (Thermo Fisher Scientific) according to the manufacturer's protocol. All mouse RNAs were analyzed on Agilent Bioanalyzer for quality assessment with RNA Integrity Number (RIN) ranging from 8.8 to 9.6 with a median of 9.5. cDNA libraries were prepared using 2 ng of total RNA and 1 μl of a 1:50,000 dilution of ERCC RNA Spike in Controls (ThermoFisher Scientific) using the SMARTSeq v2 protocol (Picelli et al, 2014) with the following modifications: (1) Addition of 20 μM template switching oligos (TSO); (2) Use of 200 pg cDNA with 1/5 reaction of Illumina Nextera XT kit (Illumina). The length distribution of the cDNA libraries was monitored using a DNA High Sensitivity Reagent Kit on the Labchip (Perkin Elmer). All samples were subjected to an indexed paired-end sequencing run of 2 × 151 cycles on an Illumina HiSeq 4000 system (~26 samples/lane).

## Bioinformatics and transcriptome analysis

STAR aligner (Dobin et al, 2013) was used for mapping paired-end raw reads to GRCm38 mouse genome build. Reads that were mapped to genes were counted using featureCounts (Liao et al, 2014) based on GENCODEvM20 gene annotation (Harrow et al, 2006). edgeR Bioconductor package (Robinson et al, 2010) was used for calculation of Log2 transformed counts per million mapped read (log2CPM) and log2 transformed reads per kilobase per million mapped reads (log2RPKM). Genes with log2CPM inter-quartile range (IQR) less than 0.5 across all samples were filtered out from subsequent differential expression gene (DEG) analysis. Differential gene expression was analyzed between groups pairwise (CD64$^+$MHCII$^+$ vs. CD64$^+$MHCII$^-$) was done using edgeR. Genes with a Benjamini-Hochberg (Benjamini and Hochberg, 1995) false discovery rate (FDR) multiple testing corrected $p$-value < 0.05 were considered differentially expressed. DEGs were subsequently uploaded into Cytoscape (version 3.9.1) (Shannon et al, 2003) querying the Gene Ontology (GO) – biological function database using plugins ClueGo (Bindea et al, 2009) and CluePedia (Bindea et al, 2013) following these parameters GO Term fusion selected; display pathways with $p$ values ≤ 0.05; GO tree intervals: all levels; GO term minimum 3 genes; threshold of 4% or genes per pathway and kappa score of 0.42.

## ELISpot assay

T-cell IFNγ ELISpot was performed as previously described (Teo et al, 2013), with some modifications listed as follows: Joint-footpad cells at 6 dpi from CHIKV-infected three-week-old animals were harvested as described above. Following, CD4$^+$ T cells were isolated via negative selection means with CD4 T cells isolation kit (MACS Miltenyi) following manufacturer's protocol. CD64$^+$MHCII$^-$ and CD64$^+$MHCII$^+$ macrophages were sorted as described above. A total of 25,000 CD4$^+$ T cells and 50,000 macrophages (either CD64$^+$MHCII$^+$ or CD64$^+$MHCII$^-$) were stimulated with 30 μg/mL CHIKV E2EP3 peptide (Kam et al, 2012) in a final volume of 100 μL complete RPMI medium containing 30 U/mL IL-2 for 15 h. For negative controls, CHIKV E2EP3 peptides were not added.

## The paper explained

### Problem

Chikungunya is a neglected tropical disease caused by infection with the chikungunya virus (CHIKV). Earlier studies in murine models have shown that CD4$^+$ T cells are drivers of the joint-footpad inflammation. Importantly, elevated numbers of inflammatory myeloid cells are also observed in the inflamed tissue, indicating their likely contribution to the disease pathogenesis.

### Results

Using murine models of CHIKV infection, we first demonstrated the presence of activated macrophages in the joint-footpad. Further investigation identify these macrophages as CD64$^+$MCII$^+$ and their recruitment and activation is dependent on GM-CSF and IFNγ secreted by activated CD4$^+$ T cells. Subsequently, transcriptomics profiling indicated their involvement in T-cell proliferation and differentiation pathways. Lastly, functional involvement of CD64$^+$MHCII$^+$ macrophages in driving CHIKV immunopathogenesis was demonstrated with clodronate-liposome depletion of macrophages 24 h before the peak of joint-footpad pathology. This resulted in significant reduction of joint-footpad inflammation.

### Impact

Our study highlights the importance of pathogenic immune-crosstalk between the CD4$^+$ T cells and CD64$^+$MHCII$^+$ macrophages in the immunopathogenesis of CHIKV infection. We believe that this study provides new insights in the disease mechanisms and provides opportunities for future development of novel host-directed immunotherapeutic drugs in combating against CHIKV and related arboviruses.

## Statistical analysis

Animals were randomized into the various experimental groups prior to start of each experiment. Comparisons between the different experimental groups or samples were performed using non-parametric Mann–Whitney rank sum test (two-tailed) or one-way ANOVA. No blinding was done. Analyses were performed with GraphPad PRISM (version 9) (GraphPad Software). *P* values of <0.05 are statistically significant.

## Data availability

Data are accessible at NCBI's Gene Expression Omnibus (GEO) database (accession number GSE208540).

## Peer review information

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

## Acknowledgements

We would like to thank Nurhidaya Binte Shadan, Ivy Low, Seri Mustafah, and Leon Hwang (SIgN flow cytometry team) and the SIgN Immunomonitoring Group for their assistance. We would also like to thank Geraldine Koh, Josephine Lum, and Shanshan Wu Howland from the SIgN Immunogenomics Platform for their assistance. We are also thankful to previous lab members Wendy Lee, Zhisheng Her, Cheryl Lee, Tze-Kwang Chua, and William Xie for the technical assistance provided. Lastly, we are grateful to Daniel Ackerman of Insight Editing London for editing the manuscript prior to submission. This work was supported by the Singapore Biomedical Research Council (BMRC; core research grants to the A*STAR Infectious Diseases Labs (ID Labs)). Fok-Moon Lum and Guillaume Carissimo are supported by the Singapore National Medical Research Council (NMRC), Open-Fund Young Investigator Research Grant (OF-YIRG) (OFYIRG22jul-0044 and OFYIRG19nov-0051, respectively). Flow cytometry platform is supported by the Health and Biomedical Sciences (HBMS) Open Fund Shared Infrastructure Support Grant under the Immunomonitoring Service Platform project (NRF2017_SISFP09).

## Author contributions

**Fok-Moon Lum**: Conceptualization; Data curation; Software; Formal analysis; Supervision; Validation; Investigation; Visualization; Methodology; Writing— original draft; Project administration; Writing—review and editing. **Yi-Hao Chan**: Conceptualization; Data curation; Software; Formal analysis; Supervision; Validation; Investigation; Methodology; Writing—original draft; Project administration; Writing—review and editing. **Teck-Hui Teo**: Conceptualization; Data curation; Formal analysis; Supervision; Validation; Methodology; Writing—original draft; Project administration; Writing—review and editing. **Etienne Becht**: Software; Formal analysis; Validation; Investigation; Methodology; Writing—original draft. **Siti Naqiah Amrun**: Data curation; Validation; Investigation; Methodology; Project administration. **Karen WW Teng**: Conceptualization; Formal analysis; Investigation; Methodology. **Siddesh V Hartimath**: Conceptualization; Data curation; Formal analysis; Validation; Investigation; Visualization; Methodology; Writing—original draft. **Nicholas KW Yeo**: Data curation; Formal analysis; Investigation; Methodology; Writing —original draft. **Wearn-Xin Yee**: Formal analysis; Investigation; Methodology. **Nicholas Ang**: Software; Formal analysis; Validation; Investigation; Methodology; Writing—original draft. **Anthony M Torres-Ruesta**: Formal analysis; Investigation; Methodology; Writing—original draft; Project administration. **Siew-Wai Fong**: Formal analysis; Validation; Investigation; Methodology; Writing—original draft; Project administration. **Julian L Goggi**: Conceptualization; Software; Formal analysis; Validation; Investigation; Visualization; Methodology; Writing—original draft. **Evan W Newell**: Conceptualization; Software; Formal analysis; Validation; Investigation; Visualization; Methodology. **Laurent Renia**: Conceptualization; Validation; Writing—original draft; Project administration. **Guillaume Carissimo**: Conceptualization; Data curation; Software; Formal analysis; Validation; Investigation; Methodology; Writing—original draft; Project administration; Writing—review and editing. **Lisa FP Ng**: Conceptualization; Resources; Data curation; Supervision; Writing—original draft; Project administration; Writing— review and editing.

## Disclosure and competing interests statement

Dr. Lisa Ng is a Member of the EMBO Molecular Medicine Editorial Board. This has no bearing on the editorial consideration of this article for publication.

# Expanded View Figures

**Figure EV1.  High-dimensional analysis of mass cytometry data and Immunophenotyping gating strategy.**

(**A**) Cluster ID annotation with heatmap. Cluster IDs are indicated in rows, while surface marker expression levels are indicated by the columns. The color represents the relative mean signal intensity of a surface marker. Blue and red represent low and high intensity, respectively. (**B**) Cellular infiltrates in the CHIKV-infected joint-footpad at 6 days post-infection (dpi) were determined with flow cytometry. Brieftly, live $CD45^+$ immune cells were gated out before the $CD3^+$ (inclusive of $CD4^+$ and $CD8^+$) T cells, $CD3^+NK1.1^+$ NKT cells, $CD11b^+Ly6G^+$ neutrophils, $NK1.1^+$ NK cells, $CD11b^+Ly6C^+$ monocytes and $CD64^+$ macrophages were identified. Plots shown are representative of a single CHIKV-infected mouse.

▶

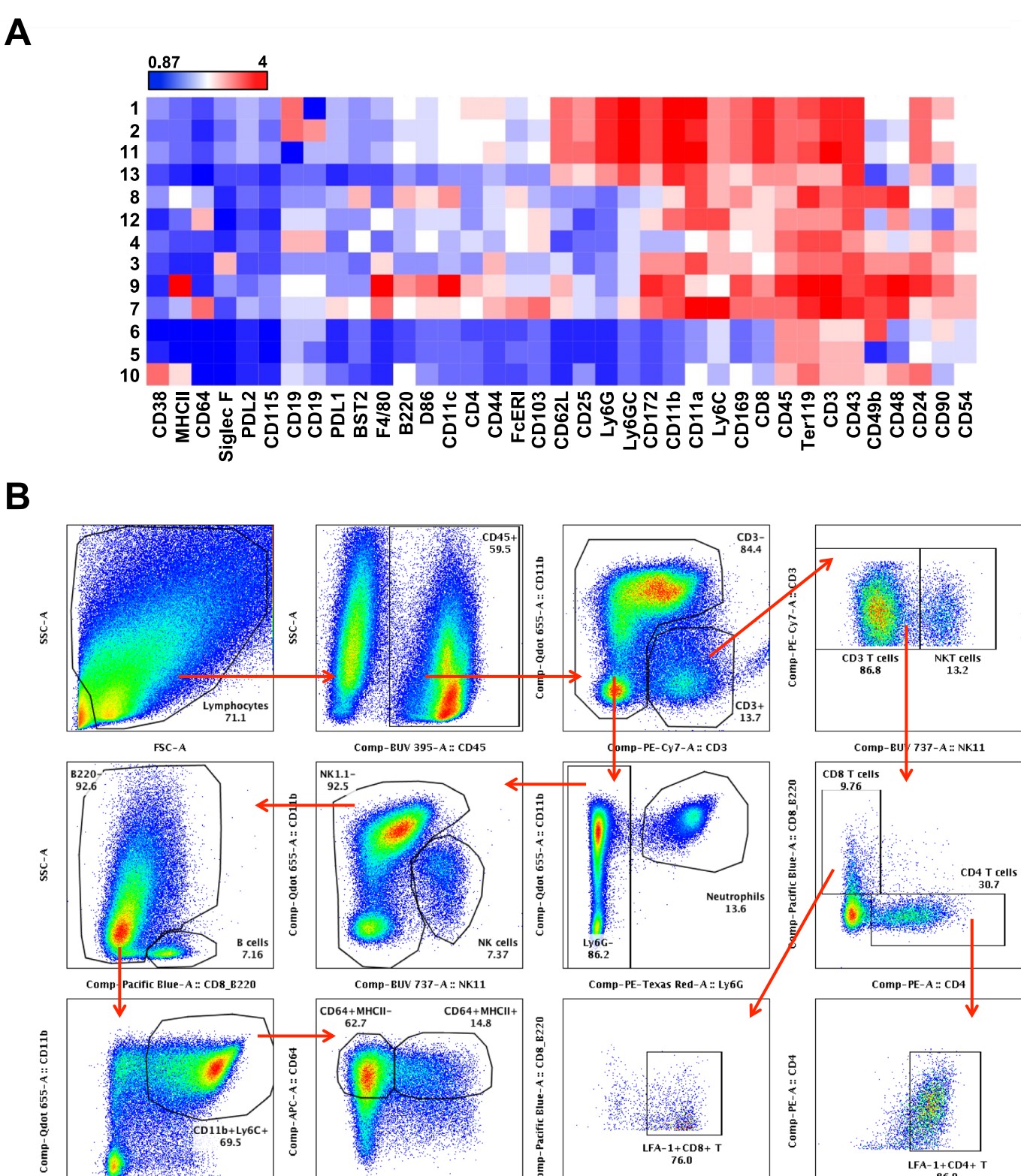

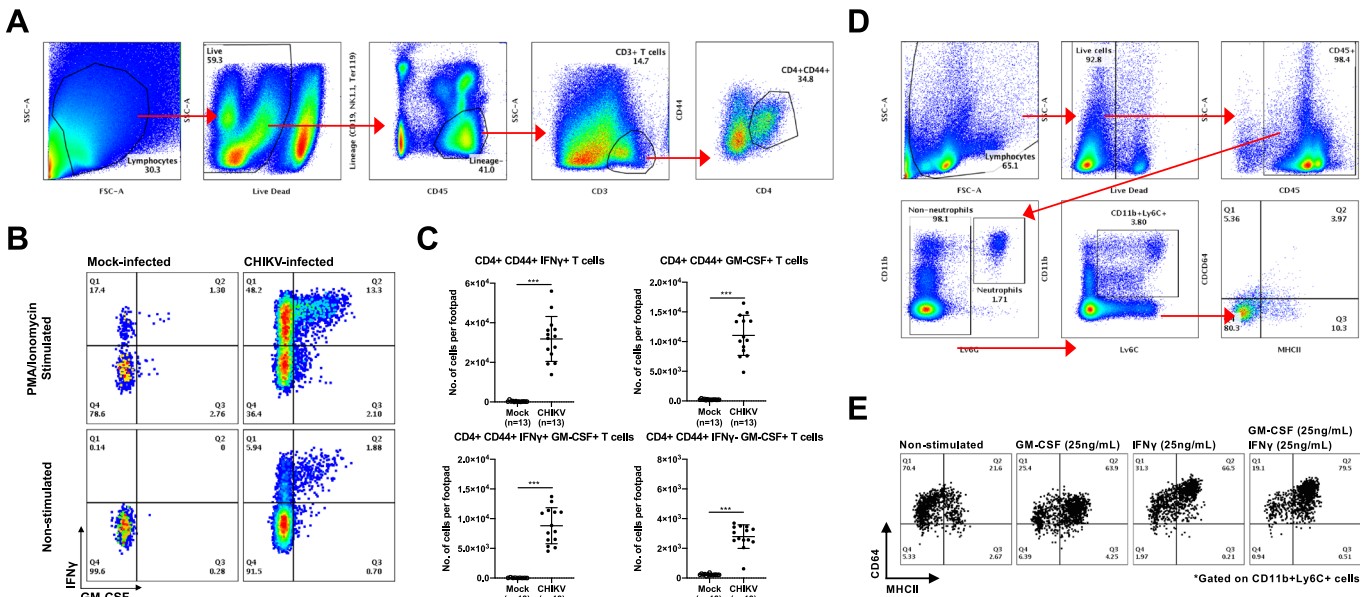

**Figure EV2. Intracellular staining reveals presence of IFNγ- and GM-CSF-producing CD4⁺CD44⁺ T cells during CHIKV infection.**

(A) Joint-footpad cells from non-infected and CHIKV-infected animals were harvested at 6 dpi and were subjected to stimulation with ionomycin and Phorbol Myristate Acetate (PMA) for 4 h. Stimulated cells were subsequently stained for the presence of GM-CSF and IFNγ in CD4⁺CD44⁺ memory/effector T cells. Representative electronic gating strategy to isolate CD4⁺CD44⁺ T cells from joint-footpad cells. Plots shown are concatenated from CHIKV-infected samples. (B) Representative flow cytometry plot illustrating the gating strategy in the identification of GM-CSF and IFNγ within the CD4⁺CD44⁺ T cells. (C) Dotplots showing the numbers of the various identified subsets within the joint-footpads. Data were from biological replicates obtained from two independent experiments. All data are presented as mean ± SD. Data comparisons between the groups were performed with non-parametric Mann–Whitney $U$ test (two-tailed). IFNγ⁺ T cells, ***$P = 0.0000000499$; GM-CSF⁺ T cells, ***$P = 0.0000000499$; IFNγ⁺GM-CSF⁺ T cells, ***$P = 0.0000000499$; IFNγ⁻GM-CSF⁺ T cells, ***$P = 0.0000000499$. (D) Fresh blood was obtained from non-infected animals. Red blood cells were subsequently lysed and live cells were stained with a commercially available live/dead dye before being stained with a cocktail of antibodies targeting surface markers CD45, CD11b, Ly6G, Ly6C, CD64, and MHCII. Live CD45⁺ cells were firstly identified, and non-neutrophils were gated next. CD11b⁺Ly6C⁺ monocytes were identified from the non-neutrophils and were shown to be low in CD64 and MHCII expression (Q4). Plots shown are representative of a single non-infected mouse. (E) Peripheral whole blood was obtained from non-infected animals and were subjected to GM-CSF and IFNγ stimulation after removal of the red blood cells. Stimulated cells were harvested 24 h later and stained for the identification of monocytes and macrophage subsets. Representative flow cytometry plots showing the gating strategy used to identify CD64⁺MHCII⁺ cells among the precursor CD11b⁺Ly6C⁺ monocytes.

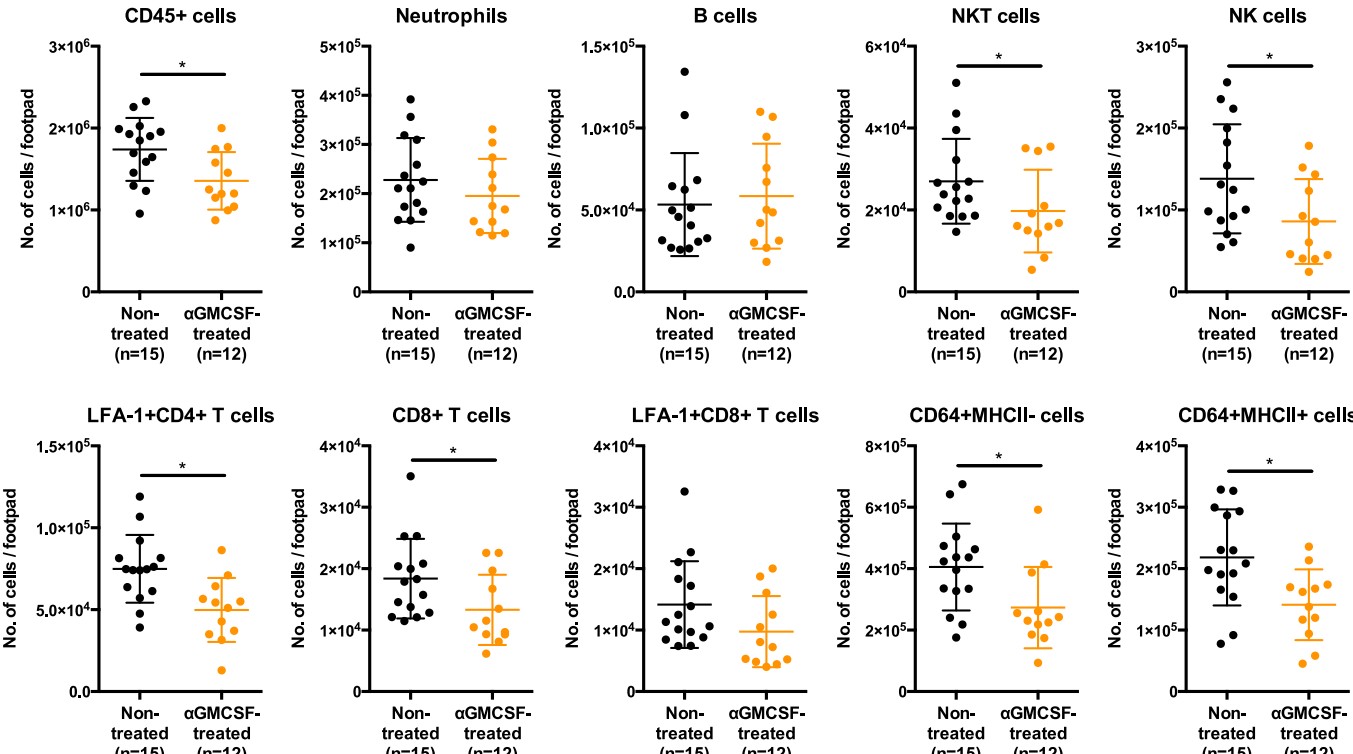

**Figure EV3.    Depletion of GM-CSF alters the numbers of immune cells infiltrating the joint-footpad.**

CHIKV infection (1 × 10⁶ PFU) was performed in the right footpad of wild-type animals. Four days post-infection (dpi), anti-GM-CSF antibodies were given intraperitoneally. Subsequently, numbers of infiltrating CD45$^+$ cells, neutrophils, B cells, NKT cells. NK cells, LFA-1$^+$CD4$^+$ T cells, CD8$^+$ T cells, LFA-1$^+$CD4$^+$ T cells, CD11b$^+$Ly6C$^+$ cells, CD64$^+$MHCII$^-$ cells and CD64$^+$MHCII$^+$ cells were obtained following immunophenotyping of the joint-footpad at 6 days post-infection (dpi). Data were from biological replicates obtained from two independent experiments. All data are presented as mean ± SD. Data comparisons between the groups were performed with non-parametric Mann–Whitney $U$ test (two-tailed): CD45 + , *$P$ = 0.0139; NKT, *$P$ = 0.0469; NK, *$P$ = 0.0214; LFA-1$^+$CD4$^+$ T, *$P$ = 0.031; CD8$^+$ T, *$P$ = 0.0281; CD64$^+$MHCII$^-$, *$P$ = 0.0186; CD64$^+$MHCI$^+$, *$P$ = 0.0161.

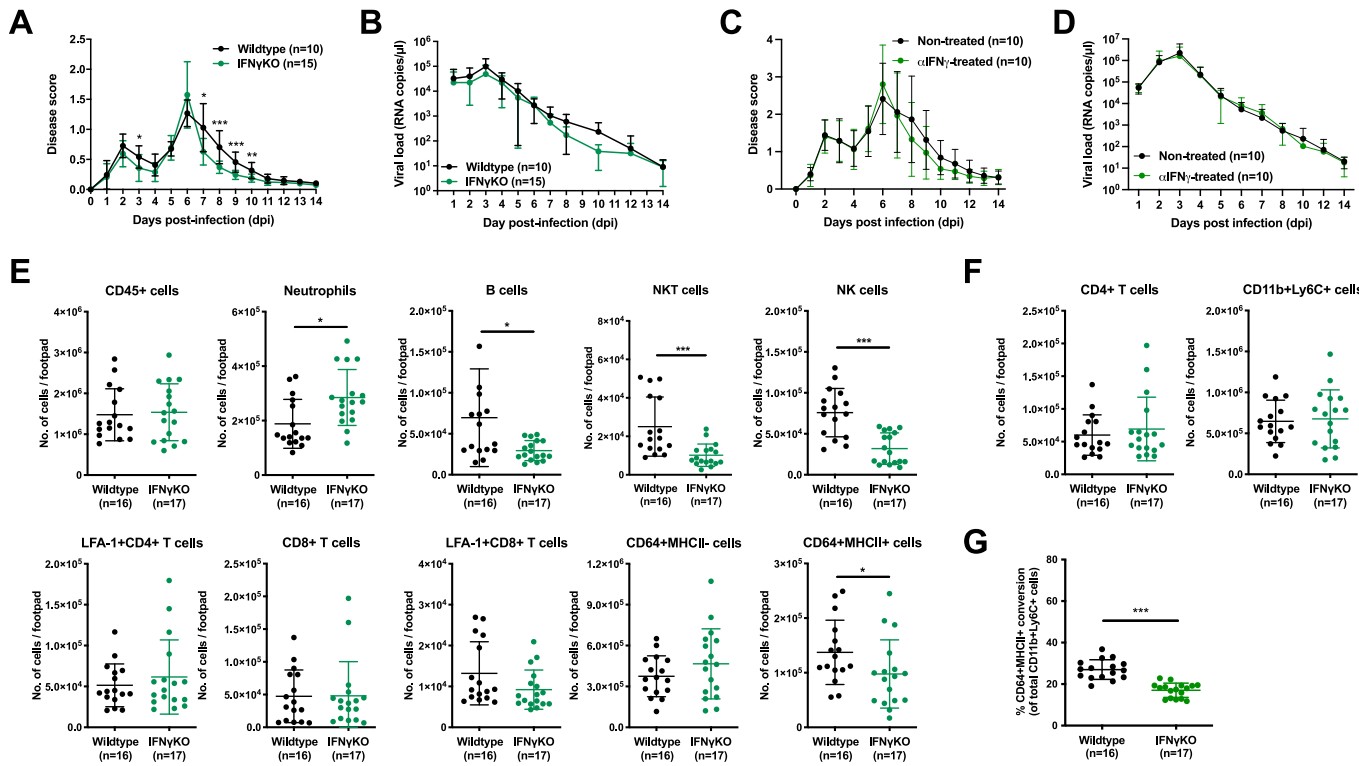

**Figure EV4. Absence of IFNγ alters the conversion of CD64⁺MHCII⁺ macrophages in the joint-footpad.**

(A,B) IFNγKO animals were infected with 1 × 10⁶ PFU of CHIKV in the right footpad. Joint-footpad swelling (**A**) and viral RNA load (**B**) were monitored over 14 days. Data were from biological replicates obtained from two independent experiments and presented as mean ± SD. Comparisons between the groups were performed with non-parametric Mann–Whitney *U* test (two-tailed). For footpad swelling: 3 dpi, \*P = 0.0395; 7 dpi, \*P = 0.0216; 8 dpi, \*\*\*P = 0.00000214; 9 dpi, \*\*\*P = 0.0003; 10 dpi, \*\*P = 0.0051. (**C,D**) Wild-type animals were infected with 1 × 10⁶ PFU of CHIKV in the right footpad and were treated with anti-IFNγ antibodies at 0-, 2- and 4-days post-infection (dpi). Joint-footpad swelling (**C**) and viral RNA load (**D**) were monitored over 14 days. Data were from biological replicates obtained from two independent experiments and presented as mean ± SD. (**E–G**) Immunophenotyping of IFNγKO joint-footpad was performed at 6 dpi to determine the numbers of infiltrating CD45⁺ cells, neutrophils, B cells, NKT cells. NK cells, LFA-1⁺CD4⁺ T cells, CD8⁺ T cells, LFA-1⁺CD8⁺ T cells, CD64⁺MHCII⁻ cells and CD64⁺MHCII⁺ cells (**E**). Numbers of joint-footpad infiltrating CD4⁺ T cells and CD11b⁺Ly6C⁺ monocytes are depicted (**F**). Percentage differentiation of CD11b⁺Ly6C⁺ monocytes into CD64⁺MHCII⁺ macrophages in CHIKV-infected wild type or IFNγKO animals is shown (**G**). Data were from biological replicates obtained from two independent experiments and presented as mean ± SD. Comparisons between the groups were performed with non-parametric Mann–Whitney *U* test (two-tailed). For neutrophils, \*P = 0.0136; B cells, \*P = 0.0136; NKT, \*\*\*P = 0.0002; NK, \*\*\*P = 0.0001; CD64⁺MHCII⁺, \*P = 0.0207; for % CD64⁺MHCII⁺ conversion (**G**), \*\*\*P = 0.000000105.

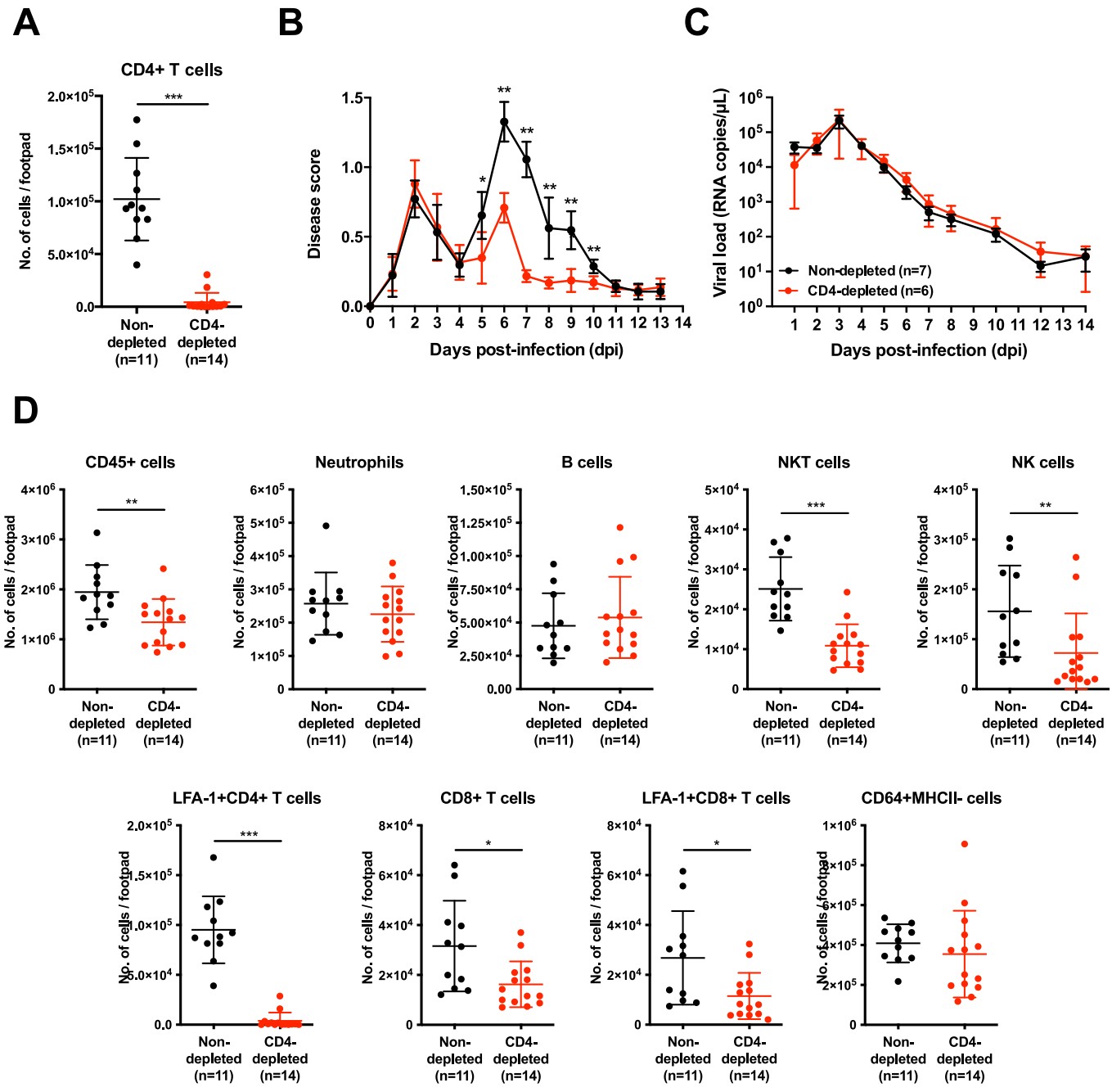

**Figure EV5. CD4-depletion reduces CHIKV disease severity.**

(A) Wild-type animals were infected with $1 \times 10^6$ PFU of CHIKV in the right footpad and anti-CD4 antibodies were given intraperitoneally at -1 and 4 days post-infection (dpi). Immunophenotyping of joint-footpad was performed at 6 dpi to determine the numbers of infiltrating CD4+ T cells. Data were from biological replicates obtained from two independent experiments and presented as mean ± SD. Data comparisons between the groups were performed with non-parametric Mann–Whitney $U$ test (two-tailed). CD4+ T cells, ***$P$ = 0.000000449. (B,C) Joint-footpad swelling (B) and viral RNA load (C) of CHIKV-infected animals with or without administration of anti-CD4 antibodies, over a period of 14 days. Data were from biological replicates obtained from two independent experiments and presented as mean ± SD. Data comparisons between the groups were performed with non-parametric Mann–Whitney $U$ test (two-tailed). For footpad swelling: 5 dpi, *$P$ = 0.0192; 6 dpi, **$P$ = 0.0012; 7 dpi, **$P$ = 0.0012; 8 dpi, **$P$ = 0.0012; 9 dpi, **$P$ = 0.0012; 10 dpi, **$P$ = 0.0023. (D) Numbers of infiltrating CD45+ cells, neutrophils, B cells, NKT cells. NK cells, LFA-1+CD4+ T cells, CD8+ T cells, LFA-1+CD4+ T cells, CD11b+Ly6C+ cells and CD64+MHCII- cells were determined following immunophenotyping of the joint-footpad at 6 dpi (A). Data were from biological replicates obtained from two independent experiments and presented as mean ± SD. Data comparisons between the groups were performed with non-parametric Mann–Whitney $U$ test (two-tailed). CD45+, **$P$ = 0.0042; NKT, ***$P$ = 0.0000306; NK, **$P$ = 0.0075; LFA-1+CD4+ T, ***$P$ = 0.000000449; CD8+, *$P$ = 0.0108; LFA-1+CD8+ T, *$P$ = 0.0179.

