## [Peer Review File · EMBO Molecular Medicine]

Crosstalk between CD64⁺MHCII⁺ macrophages and CD4⁺ T cells drives joint pathology during chikungunya

Fok-Moon Lum, Yi-Hao Chan, Teck-Hui Teo, Etienne Becht, Siti Naqiah Amrun, Karen Teng, Siddesh Hartimath, Nicholas Kim-Wah Yeo, Yee Wearn-Xin, Nicholas Ang, Anthony Torres-Ruesta, Siew-Wai Fong, Julian Goggi, Evan Newell, Laurent RENIA, Guillaume Carissimo, and Lisa Ng

Corresponding authors: Lisa Ng (lisa_ng@idlabs.a-star.edu.sg) , Fok-Moon Lum (lum_fok_moon@idlabs.a-star.edu.sg)

Review Timeline:

Submission Date:	26th Jul 23
Editorial Decision:	31st Aug 23
Revision Received:	18th Dec 23
Editorial Decision:	8th Jan 24
Revision Received:	12th Jan 24
Accepted:	17th Jan 24

Editor: Lise Roth

Transaction Report:

31st Aug 2023

Dear Dr. Ng,

Thank you for submitting your work to EMBO Molecular Medicine, and please accept my apologies for the delay in getting back to you during this busy time of the year. We have now heard back from the referees who agreed to evaluate your manuscript. As you will see below, the reviewers raise substantial concerns on your work, which unfortunately preclude its publication in EMM in its current form. The reviewers find that the question addressed by the study is of potential interest, however they remain unconvinced that some of the major conclusions are sufficiently supported by the data.

If you feel you can satisfactorily address the concerns raised by the referees, you may wish to submit a revised version of your manuscript. In particular, evidence for a direct role of CD64+MHCII+ in CHIKV arthritis pathogenesis should be strengthened.

Please attach a covering letter giving details of the way in which you have handled each of the points raised by the referees. A revised manuscript will once again be subject to review, and we cannot guarantee at this stage that the eventual outcome will be favorable.

We are expecting your revised manuscript within three months, if you anticipate any delay, please contact us.

We require:

- 1) A .docx formatted version of the manuscript text (including legends for main figures, EV figures and tables). Please make sure that the changes are highlighted to be clearly visible.
- 2) Individual production quality figure files as .eps, .tif, .jpg (one file per figure). For guidance, download the 'Figure Guide PDF' (<https://www.embopress.org/page/journal/17574684/authorguide#figureformat>).
- 3) At EMBO Press we ask authors to provide source data for the main figures. Our source data coordinator will contact you to discuss which figure panels we would need source data for and will also provide you with helpful tips on how to upload and organize the files.
- 4) A .docx formatted letter INCLUDING the reviewers' reports and your detailed point-by-point responses to their comments. As part of the EMBO Press transparent editorial process, the point-by-point response is part of the Review Process File (RPF), which will be published alongside your paper.
- 5) A complete author checklist, which you can download from our author guidelines (<https://www.embopress.org/page/journal/17574684/authorguide#submissionofrevisions>). Please insert information in the checklist that is also reflected in the manuscript. The completed author checklist will also be part of the RPF.
- 6) Please note that all corresponding authors are required to supply an ORCID ID for their name upon submission of a revised manuscript. An ORCID identifier is currently missing for Dr. Fok-Moon Lum.
- 7) It is mandatory to include a 'Data Availability' section after the Materials and Methods. Before submitting your revision, primary datasets produced in this study need to be deposited in an appropriate public database, and the accession numbers and database listed under 'Data Availability'.
- 8) For data quantification: please specify the name of the statistical test used to generate error bars and P values, the number (n) of independent experiments (specify technical or biological replicates) underlying each data point and the test used to calculate p-values in each figure legend. The figure legends should contain a basic description of n, P and the test applied. Graphs must include a description of the bars and the error bars (s.d., s.e.m.). Please provide exact p values.
- 9) Our journal encourages inclusion of *data citations in the reference list* to directly cite datasets that were re-used and obtained from public databases. Data citations in the article text are distinct from normal bibliographical citations and should directly link to the database records from which the data can be accessed. In the main text, data citations are formatted as follows: "Data ref: Smith et al, 2001" or "Data ref: NCBI Sequence Read Archive PRJNA342805, 2017". In the Reference list,

data citations must be labeled with "[DATASET]". A data reference must provide the database name, accession number/identifiers and a resolvable link to the landing page from which the data can be accessed at the end of the reference. Further instructions are available at .

10) Author contributions: CRediT has replaced the traditional author contributions section because it offers a systematic machine readable author contributions format that allows for more effective research assessment. Please remove the Authors Contributions from the manuscript and use the free text boxes beneath each contributing author's name in our system to add specific details on the author's contribution. More information is available in our guide to authors.

11) Disclosure statement and competing interests: We updated our journal's competing interests policy in January 2022 and request authors to consider both actual and perceived competing interests. Please review the policy <https://www.embopress.org/competing-interests> and update your competing interests if necessary.

12) Every published paper now includes a 'Synopsis' to further enhance discoverability. Synopses are displayed on the journal webpage and are freely accessible to all readers. They include a short stand first (maximum of 300 characters, including space) as well as 2-5 one-sentences bullet points that summarizes the paper. Please write the bullet points to summarize the key NEW findings. They should be designed to be complementary to the abstract - i.e. not repeat the same text. We encourage inclusion of key acronyms and quantitative information (maximum of 30 words / bullet point). Please use the passive voice. Please attach these in a separate file or send them by email, we will incorporate them accordingly.

13) As part of the EMBO Publications transparent editorial process initiative (see our Editorial at <http://embomolmed.embopress.org/content/2/9/329>), EMBO Molecular Medicine will publish online a Review Process File (RPF) to accompany accepted manuscripts.

In the event of acceptance, this file will be published in conjunction with your paper and will include the anonymous referee reports, your point-by-point response and all pertinent correspondence relating to the manuscript. Let us know whether you agree with the publication of the RPF and as here, if you want to remove or not any figures from it prior to publication. Please note that the Authors checklist will be published at the end of the RPF.

I look forward to receiving your revised manuscript.

Yours sincerely,

Lise Roth

Please use this link to login to the manuscript system and submit your revision:
<https://embomolmed.msubmit.net/cgi-bin/main.plex>

EMBO Press participates in many Publish and Read agreements that allow authors to publish Open Access with reduced/no publication charges. Check your eligibility: <https://authorservices.wiley.com/author-resources/Journal-Authors/open-access/affiliation-policies-payments/index.html>

***** Reviewer's comments *****

Referee #1 (Comments on Novelty/Model System for Author):

The overall experimental design is rigorous, evidenced by large animal numbers and appropriate statistical analysis. The data analysis, organization and interpretation are sound.

Although the role of macrophages in CHIKV arthritis pathogenesis has been fairly studied, the MHCII positive sub-population has not been investigated in the context of CHIKV infection.

The translational potential of inhibiting macrophages to treat CHIKV arthritis is not clear at this stage since the role of macrophages in CHIKV arthritis is yet clearly defined.

The murine model is well accepted to the CHIKV field.

Referee #1 (Remarks for Author):

Chikungunya virus (CHIKV) is a mosquito-borne virus that causes debilitating arthritogenic diseases. Immune cell infiltration is the hallmark of both acute and chronic CHIKV arthralgia. The authors have previously shown that CD4+ T cells contribute to CHIKV pathogenesis. In this manuscript, the authors describe a pathogenic role of CD64+MHCII+ macrophages in CHIKV arthritis. The authors observed infiltration of large numbers of CD64+ MHCII+ and CD64+ MHCII- macrophages in the joint-footpad, preceded by CD11b+ Ly6C+ inflammatory monocyte precursors, following CHIKV infection. They found that recruitment and differentiation of CD64+ cells were dependent on CD4+ T cells and GM-CSF. GM-CSF and type II IFN were able to induce MHCII expression in CD64+ cells. Depletion of macrophages with clodronate or neutralization of GM-CSF post CHIKV infection alleviated CHIKV pathogenesis, accompanied by reduced CD4+ T cell numbers. Conversely, depletion of CD4+ T cells reduced CD64+ MHCII+ cell numbers in feet. Finally, RNA-seq analysis revealed DEGs between CD64+ MHCII+ and CD64+ MHCII- from infected mice. The authors concluded that CD64+ MHCII+ cells synergize with CD4+ T cells to induce immunopathology during CHIKV infection. While these observations are interesting, several concerns need to be addressed.

Major points

1. The evidence (The effect of CD4+ T depletion on CD64+ MHCII+ macrophages, macrophage depletion on CD4+ T etc.) supporting CD64+ MHCII+ macrophages in CHIKV arthritis pathogenesis is largely correlative. Clodronate depletes all macrophages (Fig.6). Considering the controversial role of monocytes/macrophage in CHIKV pathogenesis, direct evidence underlying CD64+ MHCII+ cells in CHIKV immunopathology may be helpful. Adoptive transfer of MHCII+/MHCII- macrophages (Fig. 2) to macrophage-deficient mice followed by CHIKV challenge may provide direct evidence. At a minimum, the conclusions should be toned down and limitations should be discussed.
2. Fig.2 and EV4: Although IFN- γ stimulates MHCII expression in CD64+ cells, its role in CHIKV pathogenesis is inconsistent. IFN- γ was previously shown by the same group to be important for controlling CHIKV replication, not foot swelling though. CD4+ T cells contribute to CHIKV arthritis pathogenesis in an IFN- γ -independent manner (PMID 23209328). Here, IFN- γ KO mice presented similar viral loads but reduced swelling (A, B). Could the authors reconcile these discrepancies?
3. Fig.3 How about the viral loads in foot pads?
4. Fig.4 Serum GM-CSF/ IFN- γ levels should be shown. Is the total CD64+ MHCII+ macrophages/foot reduced in CD4+ T-depleted mice?
5. Depletion of GM-CSF/ IFN- γ had a very moderate impact on CD64+MHCII+ macrophage numbers in vivo (Fig.3, EV3, EV4), suggesting certain redundancy. It may be interesting to see if depletion of GM-CSF in IFN- γ KO mice alleviates CHIKV arthritis and reduces CD64+ MHCII+ macrophages much better.
6. Fig.6 A/B: Re-stimulation of CHIKV-specific CD4+ T with CHIKV antigen alone may stimulate IFN- γ production. A negative control without CD64+ macrophages should be included.

Minor points:

7. Fig.1 Please identify all the clusters in Fig. 1 C/D. It is notable that the Clusters 4, 5, 6 are reduced.
8. Fig.2 One-Way ANOVA is more appropriate for multiple group comparison.

Referee #2 (Remarks for Author):

Lum and colleagues provide evidence of a functional cross-talk between CD4 T cells and a subpopulation of macrophages in the pathophysiology of chikungunya infected mice. Authors interrogated an accepted mouse model, and performed some ex vivo work to establish whether two cytokines found in vivo and relevant in case of this infection are necessary and sufficient to

differentiate blood monocytes.

General comment.

This work builds up upon previous work of the team showcasing the role of CD4 T cells in the pathophysiology of this infection. Here, they provide some additional details including the contribution of a subpopulation of monocytes/macrophages. This team has already shown the contribution of IFN γ (Teo et al., 2013), and the relevance of CD4 T cells but CD8 T cells (Teo et al., 2013) as well as some hints of the reported cross talk reported here ((Carissimo et al., 2019; Felipe et al., 2020; Haist 121 et al., 2017; Lee et al, 2015; Miner et al., 2017; Rulli et al., 2011; Teo et al., 122 2017; Teo et al, 2015; Teo et al., 2013), reducing perhaps the originality of the work.

The in vivo work adds to the translation potential of these results.

Comments.

1. The Discussion can be shortened because there is some repetition of the results. Also, I will urge authors to include a paragraph relating the findings of this mouse-based work to the human pathology.
2. The experiments using blocking antibodies have not used the appropriate isotype control. The control shown in Appendix 1 only shows that the antibodies do not affect the pathophysiology. While valuable information, still it is not appropriate to compare the effect of the blocking antibodies to PBS control mice. The PBS treatment only controls the infection volume but not the possible effect of the presence of an antibody.
3. In the elegant CyTOF analysis reporting on CD45+ cells, authors should be in position to identify all the clusters found.
4. Authors should report the heatmap of the markers of the clusters up in the infected mice (12, 8, and 7). Also, it is equally important to report on those down in the infected mice (clusters 5, 6, 4 and 9). Which cells are included within these clusters? could they also play any role in the reported phenotype? The authors have disregarded them but it is not enough reason they are Cd11b^{low} being CD45⁺.
5. Could the authors clarify whether they have any CD4 T cells ONLY IFN γ , and ONLY GM-CSF (the plots seem to indicate so).
6. Could the authors confirm categorically that only the CD4 T cells are those of the whole system producing the indicated cytokines? An additional analysis may be required to confirm this or to show that there are other cells positive. This will not invalidate some of the work done but it will provide a more accurate picture of the complexity of the system.
7. The fact that the depletion experiments, and the antibody-based blocking experiments showed reductions in other immune populations, that could be producers of the cytokines under consideration, may cast doubts on the specificity of the in vivo findings. This is why the previous point needs to be taken into account.
8. Some additional work is needed to robustly justify the cross-talk between CD4 T cells and the macrophages. The experiments with a total IFN γ KO are sounded. However, it will be better to test IFN γ receptor KO (total and monocyte/macrophage tissue specific).
9. There is not enough evidence demonstrating a reduce recruitment of CD4 T cells in the clordronate-treated mice. A time course experiment is needed, coupled with the assessment of cytokines implicated in recruitment, and to validate that the absence of macrophages do not increase the cell death of the CD4 T cells.
10. Is the macrophage effect on the Cd4 T cells contact dependent or it is also mediated by a secreted cytokine/chemokine (it could be IFN β ; if this the case IFNAR Ko mice could be an invaluable tool).

Referee #3 (Comments on Novelty/Model System for Author):

The experimental model/systems used in the manuscript are adequate and extensively used in the field.

Referee #3 (Remarks for Author):

In the manuscript Crosstalk between CD64⁺MHCII⁺ macrophages and CD4⁺ T cells drives joint pathology during chikungunya by Lum F-M, Chan Y-H and colleagues, authors added relevant information on immunopathogenesis of CHIKV-inflammation. Authors demonstrated that interaction between CD4⁺T cells and CD11b⁺/Ly6C⁺/CD64⁺/MHCII⁺ drives CHIKV disease in mice. The experiments are well design and performed and the manuscript is easy to follow. The manuscript is interesting and relevant for the field; however, some questions should be answered by the authors before publication.

1. On the figure caption EV5: check/corrected (letter D) ...CD8⁺ T cells, LFA-1+CD4⁺T cells, CD11b+Ly6C⁺ and ... to: ...CD8⁺ T cells, LFA-1+CD8⁺T cells, CD11b+Ly6C+CD64⁺MHCII⁻ monocytes.
2. Fig.1C - Authors compared using uMAP the cluster of cells between mock- and CHIKV-infected cells. Can authors comment more on cluster 9 on mock-infected animals which "disappeared or move" to another position/cluster during CHIKV-infection. Can authors phenotypically identify these cells?
3. Authors stated (line 140-142) that cluster 7 presents high levels of CD11b and Ly6C identifying them as tissue infiltrating monocytes. However, this cluster also presents (based on figure EV1) high levels of CD3? Which proportion of the cells on this cluster are CD11b/Ly6C and CD3?
4. Authors demonstrated the role of the depletion of GM-CSF in CHIKV pathogenesis and viral clearance (Fig.3). Authors started the depletion of GM-CSF at 4 dpi and showed a reduction in joint-footpad swelling (Fig3A). However, authors indicated a reduction in viral RNA load in the first days of the infection (2-3 days) (line 207-209). Thus, this reduction in viral RNA load occurs before GM-CSF depletion which started only at 4 dpi. How authors explain this viral RNA load reduction?

5. Suggestion: Authors demonstrated the impact of CD4 depletion on immune cells infiltrates and cytokine/chemokine secretion on footpad (Fig4C). Authors indicated a reduction on secretion of some cytokines/chemokines after CD4 depletion (lines 251-256). Specifically, authors focus on reduction of IFN γ and GM-CSF (Fig4D). However, other cytokines/chemokines are reduced after CD4 depletion and could present a role in the changes in the cell migration to the footpad of CHIKV-infected mice. Then, authors should consider to shown the reduction of other cytokine/chemokines relevant for the model in a supplementary figure.

6. Authors discussed about the lack of tools to specifically depleted CD64 $^{+}$ /MHCII $^{+}$ cells. This experiment would be interest to confirm its role (together with CD4 T cells) in CHIKV-inflammation. However, would be possible to purify CD4 $^{+}$ T cells and CD64 $^{+}$ /MHCII $^{+}$ macrophages, and transfer those cells from CHIKV-infected mice to an uninfected mouse? The simple transfer of this cells to an uninfected mice would be sufficient to trigger inflammation and footpad swelling? I am not asking for these experiments, but I would like to listen the authors opinions on that matter, if possible.

7. Finally, authors show that CD11b $^{+}$ /Ly6C $^{+}$ are further classified as CD64 $^{+}$ /MHCII $^{+}$ or CD64 $^{+}$ /MHCII $^{-}$ cells in joint footpad of CHIKV-infected mice. Also, was demonstrated that CD64 $^{+}$ /MHCII $^{+}$ present antigens to CD4 $^{+}$ and interact with these T cell population to drive CHIKV pathology (line 295-296). Based on the results present throughout the manuscript, like the RNA seq of both populations of monocytes/macrophages could authors commented more on potential function of monocyte/macrophage CD64 $^{+}$ /MHCII $^{-}$ on the site of infection? Additionally, surprisingly to me, authors stated that only 14% of freshly isolated (from peripheral blood) CD11b $^{+}$ /Ly6C $^{+}$ cells express MHCII (Line 187-188). MHCII is generally used as a marker of monocytes/macrophages, thus, could authors have commented more on this low expression of MHCII on this cell population?

Referee #1 (Comments on Novelty/Model System for Author):

The overall experimental design is rigorous, evidenced by large animal numbers and appropriate statistical analysis. The data analysis, organization and interpretation are sound.

Although the role of macrophages in CHIKV arthritis pathogenesis has been fairly studied, the MHCII positive sub-population has not been investigated in the context of CHIKV infection.

The translational potential of inhibiting macrophages to treat CHIKV arthritis is not clear at this stage since the role of macrophages in CHIKV arthritis is yet clearly defined.

The murine model is well accepted to the CHIKV field.

Referee #1 (Remarks for Author):

Chikungunya virus (CHIKV) is a mosquito-borne virus that causes debilitating arthritogenic diseases. Immune cell infiltration is the hallmark of both acute and chronic CHIKV arthralgia. The authors have previously shown that CD4+ T cells contribute to CHIKV pathogenesis. In this manuscript, the authors describe a pathogenic role of CD64+MHCII+ macrophages in CHIKV arthritis. The authors observed infiltration of large numbers of CD64+ MHCII+ and CD64+ MHCII- macrophages in the joint-footpad, preceded by CD11b+ Ly6C+ inflammatory monocyte precursors, following CHIKV infection. They found that recruitment and differentiation of CD64+ cells were dependent on CD4+ T cells and GM-CSF. GM-CSF and type II IFN were able to induce MHCII expression in CD64+ cells. Depletion of macrophages with clodronate or neutralization of GM-CSF post CHIKV infection alleviated CHIKV pathogenesis, accompanied by reduced CD4+ T cell numbers. Conversely, depletion of CD4+ T cells reduced CD64+ MHCII+ cell numbers in feet. Finally, RNA-seq analysis revealed DEGs between CD64+ MHCII+ and CD64+ MHCII- from infected mice. The authors concluded that CD64+ MHCII+ cells synergize with CD4+ T cells to induce immunopathology during CHIKV infection. While these observations are interesting, several concerns need to be addressed.

Response: We thank the reviewer for the candid comments. We have addressed the pointers raised in our replies below.

Major points

1. The evidence (The effect of CD4+ T depletion on CD64+ MHCII+ macrophages, macrophage depletion on CD4+ T etc.) supporting CD64+ MHCII+ macrophages in CHIKV arthritis pathogenesis is largely correlative. Clodronate depletes all macrophages (Fig.6). Considering the controversial role of monocytes/macrophage in CHIKV pathogenesis, direct evidence underlying CD64+ MHCII+ cells in CHIKV immunopathology may be helpful. Adoptive transfer of MHCII+/MHCII- macrophages (Fig. 2) to macrophage-deficient mice followed by CHIKV challenge may provide direct evidence. At a minimum, the conclusions should be toned down and limitations should be discussed.

Response: We wish to emphasize that current research tools allowing for a complete depletion of the CD64+MHCII+ macrophages remain limited. We did not perform the adoptive transfer of MHCII+ or MHCII- macrophages into macrophage-deficient mice, as genetically modified animals lacking macrophages experience significant physiological differences (Gordon and Luisa., *Pflugers Arch Pflug Arch Eur J Phy*, 2017), potentially affecting their responses to CHIKV pathogenesis. However, we agree with the review's view and have toned down on our claims and further discussed the limitations of our study on lines 386-405

to read "While we are limited by the lack of research tools in effectively depleting away the CD64⁺MHCII⁺ macrophages from CHIKV-infected animals, clodronate-liposome treatment at 5 dpi significantly reduced joint-footpad swelling, coupled with a reduced presence of both CD64⁺ macrophages and CD4⁺ T cells (Figures 6C and 6D). The reduced CD4⁺ T cell numbers could be attributed to their reduced induction and expansion, following the depletion of macrophages, rather than a decreased T cell infiltration given that the levels of IP-10 (Fig 6E) were not affected by clodronate-liposome treatment.

The use of clodronate-liposome is a common tool used to investigate *in vivo* functions of macrophages (Nguyen *et al*, 2021). Here, clodronate was specifically given at 5 dpi to minimize any impact from phagocytic cell absence during innate and early adaptive immune responses. Depletion just before the peak of CHIKV disease, when CD64⁺ macrophage infiltration and differentiation is high, demonstrated indirectly the importance of *in vivo* CD64⁺MHCII⁺ macrophages in mediating the priming of CD4⁺ T cells during CHIKV immunopathogenesis. This particular approach, due to the lack of better tools, was aimed to indirectly correlate the importance of macrophages in mediating CHIKV pathology along with the CD4⁺ T cells. The presence of such a concerted effort between MHCII⁺ macrophages and CD4⁺ T cells has been reported in adipose tissue meta-inflammation (Cho *et al*, 2014)."

- Gordon, Siamon, and Luisa Martinez-Pomares. "Physiological roles of macrophages." *Pflügers Archiv-European Journal of Physiology* 469.3-4 (2017): 365-374.

2. Fig.2 and EV4: Although IFN- γ stimulates MHCII expression in CD64⁺ cells, its role in CHIKV pathogenesis is inconsistent. IFN- γ was previously shown by the same group to be important for controlling CHIKV replication, not foot swelling though. CD4⁺ T cells contribute to CHIKV arthritis pathogenesis in an IFN- γ -independent manner (PMID 23209328). Here, IFN- γ KO mice presented similar viral loads but reduced swelling (A, B). Could the authors reconcile these discrepancies?

Response: We wish to explain that in the report by Teo *et al.*, (*J Immunol*, 2013), animals used were of age 6-weeks old, whereas in this manuscript, animals used were 3-4 weeks old. This could have resulted in the differences between the two studies. Furthermore, in the report by Teo *et al.*, (*J Immunol*, 2013), pathogenic CD4⁺ T cells do not mediate inflammation via IFN γ -mediated pathway. This conclusion was made in reference to the disease severity as we saw minimal differences in terms of the disease severity between the wild-type and IFN γ ^{-/-} animals. This observation was once again demonstrated in the current submitted Expanded View Figure 4. However, in this manuscript, we are now showing that during CHIKV-induced inflammation, IFN γ functions via the conversion of CD11b+Ly6C+ infiltrating monocytes into CD64+MHCII+ macrophages as demonstrated in the current submitted Figure 2.

- Teo, Teck-Hui, *et al.* "A pathogenic role for CD4⁺ T cells during Chikungunya virus infection in mice." *The Journal of Immunology* 190.1 (2013): 259-269.

Likewise, in a report by Wilson *et al.*, (*PLoS Pathog*, 2017), CHIKV-infected IFN γ ^{-/-} animals, also exhibited a marginal decrease in the disease severity during the acute phase of the disease, with no differences in viremia clearance, corroborating with our data presented in Expanded View Figure 4A and 4B. Nevertheless, Wilson *et al.*, (*PLoS Pathog*, 2017) further highlighted that the limited effects of IFN γ deficiency could be explained by the redundancy in the induction of Type II IFN (IFN γ) regulated genes.

- Wilson *et al.* "RNA-Seq analysis of chikungunya virus infection and identification of granzyme A as a major promoter of arthritic inflammation." *PLoS Pathog* 13.2 (2017): e1006155.

3. Fig.3 How about the viral loads in foot pads?

Response: Thank you for this comment. Based on our earlier publication by Teo et al., (*J Immunol*, 2013), we showed that depletion of CD4+ T cells had no effect on the viral burden in the joint-footpad. In fact, clearance of virus is shown to be mediated by B cells and antibodies (Lum et al., *J Immunol*, 2013).

- Teo et al. "A pathogenic role for CD4+ T cells during Chikungunya virus infection in mice." *J Immunol* 190.1 (2013): 259-269.
- Lum et al. "An essential role of antibodies in the control of Chikungunya virus infection." *J Immunol* 190.12 (2013): 6295-6302.

4. Fig.4 Serum GM-CSF/ IFN- γ levels should be shown. Is the total CD64+ MHCII+ macrophages/foot reduced in CD4+ T-depleted mice?

Response: Serum GM-CSF and IFN γ levels were already depicted in Figure 4D. The total number of CD64+MHCII+ macrophages were indeed significantly reduced in the footpad of CD4+ T-cell depleted animals as shown in Figure 4B. Figure 4 is attached below for reference.

Figure 4: CD4⁺ T cells depletion alters levels of critical immune mediators in the CHIKV-infected joint-footpad.

(A-B) Wildtype animals were infected with 1×10^6 PFU of CHIKV in the right footpad, and anti-CD4 antibodies were given intraperitoneally at -1 and 4 days post-infection (dpi). Immunophenotyping was performed at 6 dpi to determine the numbers of infiltrating CD11b⁺Ly6C⁺ monocytes in CD4-depleted joint-footpads (A). Percentage differentiation of CD11b⁺Ly6C⁺ monocytes into CD64⁺MHCII⁺ and CD64⁺MHCII⁻ macrophages and absolute counts of CD64⁺MHCII⁺ macrophages in CHIKV-infected non-CD4-depleted or CD4-depleted animals (B). Data presented in (A-B) were obtained from two independent experiments.

(C-D) Joint lysates were obtained from CHIKV-infected CD4-depleted, infected wildtype (non-CD4-depleted) mice, and mock-infected control mice at peak chikungunya joint pathology (6 dpi). A multiplex microbead-based assay was used to quantify the levels of immune mediators present in these samples. Heatmap showing the levels of the analyzed immune mediators in each group of animals (C). Dot plots showing the absolute quantities of IFN γ and GM-CSF, highlighting the significant differences between the various groups of animals (D). All data in (A-D) are presented in as mean \pm SD. Data comparisons between the groups were performed with non-parametric Mann-Whitney U test (two-tailed; * $p < 0.05$; ** $p < 0.01$; *** $p < 0.001$). N.D. refers to not detectable.

5. Depletion of GM-CSF/ IFN- γ had a very moderate impact on CD64+MHCII+ macrophage

numbers in vivo (Fig.3, EV3, EV4), suggesting certain redundancy. It may be interesting to see if depletion of GM-CSF in IFN- γ KO mice alleviates CHIKV arthritis and reduces CD64+MHCII+ macrophages much better.

Response: In our results we showed that both depletion of GM-CSF and absence of IFN γ significantly affected the total numbers of CD64+MHCII+ macrophages in the joint footpad. We agree that depletion of GM-CSF in IFN $\gamma^{-/-}$ animals may further reduce CD64+MHCII+ macrophages presence and thus achieved greater reduction in CHIKV joint-footpad swelling. However, we did not perform this experiment, as our objective was to demonstrate the individual effects of GM-CSF and IFN γ .

6. Fig.6 A/B: Re-stimulation of CHIKV-specific CD4+ T with CHIKV antigen alone may stimulate IFN- γ production. A negative control without CD64+ macrophages should be included.

Response: We wish to highlight that the negative control in this figure would be the CD64+MHCII- macrophages, which as shown in Figure 6A and 6B (shown below), did not lead to the secretion of IFN γ by the CD4+ T cells.

Figure 6: Crosstalk between CD64⁺MHCII⁺ macrophages and CD4⁺ T cells drive CHIKV pathogenesis.

(A-B) CD64⁺MHCII⁺ and CD64⁺MHCII⁻ macrophages were sorted and CHIKV-specific CD4⁺ T cells were isolated from joint footpad of CHIKV-infected animals at 6 dpi. CHIKV-specific CD4⁺ T cells were subsequently restimulated with CHIKV antigen in the presence of either CD64⁺MHCII⁺ or CD64⁺MHCII⁻ macrophages for 18 hours. Representative ELISpot images obtained from the two sample groups (A), Paired line graph showing the numbers of IFN γ -producing CHIKV-specific CD4⁺ T cells post-re-stimulation (B).

Minor points:

7. Fig.1 Please identify all the clusters in Fig. 1 C/D. It is notable that the Clusters 4, 5, 6 are reduced.

Response: We have since identified the different clusters and grouped them accordingly. This is reflected in the new Figure 1C (see below). Clusters 4, 5 and 6 were reduced in our data and these likely corresponds to the innate lymphoid cells, including the NK cells. However, it is important to note that the UMAP plots presented in Figure 1C, were plotted using downsampled concatenated samples (5000 cells/events per sample), providing us a visual interpretation of the cluster abundance and for easy identification of cluster-abundance difference between the samples. The UMAP plots do not represent the quantity

of cells present in the samples. We have since made changes in the main manuscript to clearly indicate this on lines 133-134 to read “The UMAP plots were normalized to 5000 cells per sample, and represent the abundance, but not the quantity of cells present in the joints.”

In Figure 1D, cluster abundance is plotted using the Z-score transformed median number of cells within each identified clusters is plotted in a heatmap. Notably, as this is still plotted using abundance levels of the defined clusters, increased abundance of clusters 7, 8 and 12 led to the “decreased” abundance of other clusters accordingly. This does not equate to a reduced absolute quantity of these cells in the infected joint-footpad. In fact, elevated numbers of immune cells in the CHIKV-infected joint-footpad have been reported previously (Teo et al., *J Immunol*, 2013; Teo et al., *J virol*, 2015; Lee et al., *J virol*, 2015).

- Teo et al. "A pathogenic role for CD4+ T cells during Chikungunya virus infection in mice." *J Immunol* 190.1 (2013): 259-269.
- Teo, Teck-Hui, et al. "Caribbean and La Reunion chikungunya virus isolates differ in their capacity to induce proinflammatory Th1 and NK cell responses and acute joint pathology." *Journal of virology* 89.15 (2015): 7955-7969.
- Lee, Wendy WL, et al. "Expanding regulatory T cells alleviates chikungunya virus-induced pathology in mice." *Journal of virology* 89.15 (2015): 7893-7904.

Figure 1C: Joint-footpad cells from CHIKV-infected and non-infected animals (n=3 per group) were harvested at 6 dpi and stained with a panel of antibodies targeting myeloid cell surface markers. Acquisition was performed with CyTOF and data were analyzed with dimension reduction technique Uniform Manifold Approximation and Projection (UMAP). Superimposed PhenoGraphs of UMAP transformed CyTOF data from mock and CHIKV-infected joints. Presence of cluster 7 as enclosed by red circle

8. Fig.2 One-Way ANOVA is more appropriate for multiple group comparison.

Response: We have repeated the analysis with One-Way ANOVA as stated in the revised Figure 2 legend.

Referee #2 (Remarks for Author):

Lum and colleagues provide evidence of a functional cross-talk between CD4 T cells and a subpopulation of macrophages in the pathophysiology of chikungunya infected mice. Authors interrogated an accepted mouse model, and performed some *ex vivo* work to establish whether two cytokines found *in vivo* and relevant in case of this infection are necessary and sufficient to differentiate blood monocytes.

Response: We thank the reviewer for this comment and we have now responded to the pointers raised.

General comment.

This work builds up upon previous work of the team showcasing the role of CD4 T cells in the pathophysiology of this infection. Here, they provide some additional details including the contribution of a subpopulation of monocytes/macrophages. This team has already shown the contribution of IFN γ (Teo *et al.*, 2013), and the relevance of CD4 T cells but CD8 T cells (Teo *et al.*, 2013) as well as some hints of the reported cross talk reported here ((Carissimo *et al.*, 2019; Felipe *et al.*, 2020; Haist 121 *et al.*, 2017; Lee *et al.*, 2015; Miner *et al.*, 2017; Rulli *et al.*, 2011; Teo *et al.*, 122 2017; Teo *et al.*, 2015; Teo *et al.*, 2013), reducing perhaps the originality of the work. The *in vivo* work adds to the translation potential of these results.

Response: We agree with the translational potential reported in our work, that macrophages would be a plausible immunotherapeutic target in battling against CHIKV infection.

Comments:

1. The Discussion can be shortened because there is some repetition of the results. Also, I will urge authors to include a paragraph relating the findings of this mouse-based work to the human pathology.

Response: We have tried our best to shorten our discussion retain only the most important information. We have also included a paragraph to discuss the role of human CD4⁺ T cells and macrophages in CHIKV immunopathogenesis. This can be found on lines 424-438 to read "In humans, CD4⁺ T cells are detected alongside the CD8⁺ T cells in the synovial and muscle biopsies of patients in the chronic phase of the disease (Hoarau *et al.*, 2010; Ozden *et al.*, 2007). Particularly, the CD4⁺ T cells were postulated to induce inflammation through the production of proinflammatory cytokines (Hoarau *et al.*, 2010; Petitdemange *et al.*, 2015). However, the exact roles of human CD4⁺ T cells in CHIKV immunopathogenesis remains under-explored (Mapalagamage *et al.*, 2022). On the other hand, macrophages were similarly identified in synovial and muscle biopsies of patients (Hoarau *et al.*, 2010; Petitdemange *et al.*, 2015). In fact, macrophages and monocytes are known target cells of CHIKV (Her *et al.*, 2010) and has been suggested to be a reservoir for chronic infection in humans (Fox & Diamond, 2016). Studies using relevant human monocytic cell lines (Felipe *et al.*, 2020; Guerrero-Arguero *et al.*, 2020; Srivastava *et al.*, 2023), as well as with primary human monocytes-derived macrophages (Lau *et al.*, 2023), reported a pro-inflammatory immune response following CHIKV infection."

2. The experiments using blocking antibodies have not used the appropriate isotype control. The control shown in Appendix 1 only shows that the antibodies do not affect the pathophysiology. While valuable information, still it is not appropriate to compare the effect of the blocking antibodies to PBS control mice. The PBS treatment only controls the infection volume but not the possible effect of the presence of an antibody.

Response: We noted on this. However, as what was being explained in the submitted manuscript, we wanted the comparison to be done against CHIKV-infected animals that did not have any other compounds introduced exogenously.

3. In the elegant CyTOF analysis reporting on CD45+ cells, authors should be in position to identify all the clusters found.

Response: Yes, as with our response to Reviewer 1 above, we have since identified the different clusters and grouped them accordingly, as reflected in the revised Figure 1C attached below.

Figure 1C: Joint-footpad cells from CHIKV-infected and non-infected animals (n=3 per group) were harvested at 6 dpi and stained with a panel of antibodies targeting myeloid cell surface markers. Acquisition was performed with CyTOF and data were analyzed with dimension reduction technique Uniform Manifold Approximation and Projection (UMAP). Superimposed PhenoGraphs of UMAP transformed CyTOF data from mock and CHIKV-infected joints. Presence of cluster 7 as enclosed by red circle

4. Authors should report the heatmap of the markers of the clusters up in the infected mice (12, 8, and 7). Also, it is equally important to report on those down in the infected mice (clusters 5, 6, 4 and 9). Which cells are included within these clusters? could they also play any role in the reported phenotype? The authors have disregarded them but it is not enough reason they are Cd11blow being CD45+.

Response: The heatmap of markers is included in Figure EV1A. Here we can easily identify the markers that are present in the different clusters. Again, as our response to Reviewer 1, uMAP plots presented in Figure 1C, were plotted using downsampled concatenated samples (5000 events/cells per sample), allowing for easy identification of cluster-abundance differences based on their relative proportion within the entire sample. The UMAP plots do not represent the quantity of cells present in the samples. We have since made changes in the main manuscript to indicate this clearly on lines 133-134 to read “The UMAP plots were normalized to 5000 cells per sample, and represent the abundance, but not the quantity of cells present in the joints.”

In Figure 1D, the Cluster Abundance is plotted using the Z-score transformed median number of cells within each identified clusters is plotted. This is still based on the relative proportion of the identified clusters within the sample. As abundance of clusters 7, 8 and 12

were increased, other clusters had to “decreased” in abundance accordingly. This does not equate to a reduced absolute quantity of these cells in the infected joint-footpad.

Clusters 4, 5 and 6 are identified as innate lymphoid cells, including NK cells. In fact, we know that absolute NK-cell numbers are increased in the joint footpad, following CHIKV infection and is known to play a role in driving the disease pathology (Teo et al., *J virol*, 2015).

- Teo, Teck-Hui, et al. "Caribbean and La Reunion chikungunya virus isolates differ in their capacity to induce proinflammatory Th1 and NK cell responses and acute joint pathology." *Journal of virology* 89.15 (2015): 7955-7969.

As our focus in this manuscript is on the myeloid cells, we are interested in CD11b+ cells and we have now stated this clearly on lines 128-129. Thus, we disregarded the CD11b- or CD11blow clusters 4, 5 and 6. Cluster 9 is defined as CD11c^{hi} DCs, which could have migrated to the draining lymph following infection.

5. Could the authors clarify whether they have any CD4 T cells ONLY IFN γ , and ONLY GM-CSF (the plots seem to indicate so).

Response: Yes, in Expanded View Figure 2D and 2C, we showed that there were indeed CD4+ T cells that secrete only IFN γ or GM-CSF. See below attached figure for reference.

Figure EV2: Intracellular staining reveals presence of IFN γ - and GM-CSF-producing CD4⁺CD44⁺ T cells during CHIKV infection.

(B) Representative flow cytometry plot illustrating the gating strategy in the identification of GM-CSF and IFN γ within the CD4⁺CD44⁺ T cells.

(C) Dotplots showing the numbers of the various identified subsets within the joint-footpads. Data presented were obtained from two independent experiments (n=13 per group). All data are presented as mean \pm SD. Data comparisons between the various groups were performed with non-parametric Mann-Whitney *U* test (two-tailed; ****p* < 0.001).

6. Could the authors confirm categorically that only the CD4 T cells are those of the whole system producing the indicated cytokines? An additional analysis may be required to confirm this or to show that there are other cells positive. This will not invalidate some of the work done but it will provide a more accurate picture of the complexity of the system.

Response: We wish to bring to the attention to data presented in Figure 4C and 4D? While the CD4+ T cells are shown here to be major producers of the GM-CSF and IFN γ , we do not believe that they are the sole producers of these cytokines. When CD4+ T cells were depleted, there are some low levels of these cytokines being detected, which could have been produced by other cell types. Nevertheless, we did show that CD4+ T cells had a dominant role in the production of GM-CSF and IFN γ in the infected joint-footpad, which is the objective of this experiment.

7. The fact that the depletion experiments, and the antibody-based blocking experiments showed reductions in other immune populations, that could be producers of the cytokines under consideration, may cast doubts on the specificity of the in vivo findings. This is why the previous point needs to be taken into account.

Response: We agree with the comments provided. Depletion or removal of a single immune mediator or a particular immune population could cause changes in other immune populations or cytokines environment, and this is a known phenomenon. However, it is beyond the scope of this study to phenotypically characterize the changes in each immune population following a depletion experiment. Importantly, we showed in this study that CD4+ T cell depletion, caused a massive reduction in numerous cytokines (Figure 4C). While we did show that CD4+ T cells are capable of secreting GM-CSF and IFN γ (Expanded View Figure 2B and 2C), and their depletion reduces that the overall levels of GM-CSF and IFN γ in the infected joint-footpad.

8. Some additional work is needed to robustly justify the cross-talk between CD4 T cells and the macrophages. The experiments with a total IFN γ KO are sounded. However, it will be better to test IFN γ receptor KO (total and monocyte/macrophage tissue specific).

Response: We wish to re-emphasize that we did show the effects of IFN γ on promoting the differentiation of CD64+MHCII+ macrophages in Figure 2. Furthermore, in CHIKV-infected IFN γ ^{-/-} animals, we also observed a reduced capacity of CD11b+Ly6C+ proinflammatory monocytes differentiating into CD64+MHCII+ macrophages (see below).

Figure: Reduced capacity to differentiate into CD64+MHCII+ macrophages in the absence of IFN γ .

(A) Immunophenotyping of IFN γ ^{-/-} joint-footpad was performed at 6 dpi to determine the numbers of numbers of joint-footpad infiltrating CD11b+Ly6C+ monocytes.

(B) Percentage differentiation of CD11b+Ly6C+ monocytes into CD64+MHCII+ macrophages in CHIKV-infected wildtype or IFN γ ^{-/-} animals is shown.

(C) Numbers of CD64+MHCII+ macrophages in the joint-footpad.

Moreover, we showed that isolated CD64⁺MHCII⁺ macrophages are strong antigen-presenting cells that could activate CD4⁺ T cells in an *ex vivo* CHIKV IFN γ -ELISpot assay. Figure 6A and 6B are attached below for easy reference.

With current limited tools, we believed that we have demonstrated the crosstalk between the CD4⁺ T cells and CD64⁺MHCII⁺ macrophages with the above well thought-out experiments presented in this study.

Figure 6: Crosstalk between CD64⁺MHCII⁺ macrophages and CD4⁺ T cells drive CHIKV pathogenesis.

(A-B) CD64⁺MHCII⁺ and CD64⁺MHCII⁻ macrophages were sorted and CHIKV-specific CD4⁺ T cells were isolated from joint footpad of CHIKV-infected animals at 6 dpi. CHIKV-specific CD4⁺ T cells were subsequently restimulated with CHIKV antigen in the presence of either CD64⁺MHCII⁺ or CD64⁺MHCII⁻ macrophages for 18 hours. Representative ELISpot images obtained from the two sample groups (A), Paired line graph showing the numbers of IFN γ -producing CHIKV-specific CD4⁺ T cells post-re-stimulation (B). Data comparisons between the groups were performed with non-parametric Mann-Whitney *U* test (two-tailed; ***p* < 0.01; ****p* < 0.001).

9. There is not enough evidence demonstrating a reduce recruitment of CD4 T cells in the clordronate-treated mice. A time course experiment is needed, coupled with the assessment of cytokines implicated in recruitment, and to validate that the absence of macrophages do not increase the cell death of the CD4 T cells.

Response: Our results presented in Figure 6D clearly shows a reduced numbers of CD4⁺ T cells present in the joint-footpad. This effect could arise from either reduced recruitment or reduced CD4⁺ T cells numbers. Since clodronate is known to deplete away all phagocytic cells (Zhang et al., *J Immunol*, 2013), including macrophages, it is plausible that clodronate treatment prevented the induction of pathogenic CD4⁺ T cells and their expansion. As such, in order to circumvent or reduce the likelihood of this, we attempted to give clodronate liposome to the infected animals only at 5 dpi, a day before peak of CHIKV joint-footpad pathology. This is to minimize any impact of phagocytic cell absence during innate and early adaptive immune responses.

- Zhang, Yi, et al. "APCs in the liver and spleen recruit activated allogeneic CD8⁺ T cells to elicit hepatic graft-versus-host disease." *The Journal of Immunology* 169.12 (2002): 7111-7118.

Nevertheless, we have also quantified the levels of IP-10, a key cytokine involved in the recruitment of T cells (Khan et al., *Immunity*, 2000) in the joint footpad. IP-10 signalling is

reported to drives pathogenesis of arthritogenic alphaviruses (Lin et al., *Viruses*, 2020). We found that there were no differences between the control group versus the clodronate-liposome treated group (see figure below). These results indicated that the recruitment of CD4+ T cells into the joint-footpad is unlikely to be affected.

- Khan, Imtiaz A., et al. "IP-10 is critical for effector T cell trafficking and host survival in *Toxoplasma gondii* infection." *Immunity* 12.5 (2000): 483-494.
- Lin, Tao, et al. "CXCL10 signaling contributes to the pathogenesis of arthritogenic alphaviruses." *Viruses* 12.11 (2020): 1252.

Figure. Quantification of IP-10 in CHIKV-infected joint-footpad following clodronate liposome treatment.

Wildtype animals were infected with 1×10^6 PFU of CHIKV in the right footpad. At 5dpi, animals were given intraperitoneally with either clodronate liposome (1mg) or empty liposome (control). Levels of IP-10 present in joint-footpad was quantified at 6 dpi for both groups of animals.

In fact, our data further demonstrated the importance of CD64+MHCII+ macrophages, as their depletion could have resulted in a potential loss of CD4+ T cell expansion in the joint-footpad. We will now incorporate this piece of new data into the revised Figure 6 (as attached below) to further explain this phenomenon on lines 390-394 to read "The reduced CD4+ T cell numbers could be attributed to their reduced induction and expansion, following the depletion of macrophages, rather than a decreased T cell infiltration given that the levels of IP-10 (Fig 6E) were not affected by clodronate-liposome treatment."

Figure 6: Crosstalk between CD64+MHCII+ macrophages and CD4+ T cells drive CHIKV pathogenesis.

(A-B) CD64⁺MHCII⁺ and CD64⁺MHCII⁻ macrophages were sorted and CHIKV-specific CD4⁺ T cells were isolated from joint footpad of CHIKV-infected animals at 6 dpi. CHIKV-specific CD4⁺ T cells were subsequently restimulated with CHIKV antigen in the presence of either CD64⁺MHCII⁺ or CD64⁺MHCII⁻ macrophages for 18 hours. Representative ELISpot images obtained from the two sample groups (A), Paired line graph showing the numbers of IFN γ -producing CHIKV-specific CD4⁺ T cells post-re-stimulation (B).

(C) Wildtype animals were infected with 1×10^6 PFU of CHIKV in the right footpad. At 5dpi, animals were given intraperitoneally with either clodronate liposome (1mg) or empty liposome (control). Joint-footpad swelling of these animals were monitored over a period of 10 days.

(D-E) Wildtype animals were infected with 1×10^6 PFU of CHIKV in the right footpad. At 5dpi, animals were given intraperitoneally with either clodronate liposome (1mg) or empty liposome (control). Immunophenotyping was performed at 6 dpi for both groups of animals. Graphs show the numbers of CD4⁺ T cells, CD11b⁺Ly6C⁺ precursor cells, CD64⁺MHCII⁻ and CD64⁺MHCII⁺ macrophages present in the joint-footpad (D). Levels of IP-10 present in joint-footpad was quantified at 6 dpi for both groups of animals (E). All data are presented as mean \pm SD. All data presented in (C-E) were obtained from two independent experiments. Data comparisons between the groups were performed with non-parametric Mann-Whitney *U* test (two-tailed; ***p* < 0.01; ****p* < 0.001).

10. Is the macrophage effect on the Cd4 T cells contact dependent or it is also mediated by a secreted cytokine/chemokine (it could be IFN β ; if this the case IFNAR Ko mice could be an invaluable tool).

Response: We hypothesize that it is contact dependent through antigen-presentation via the MHCII surface molecule. As shown in the IFN γ -ELISpot assay presented in Figure 6A and 6B, CD64⁺MHCII⁺ macrophages were able to activate the CD4⁺ T cells, but not the CD64⁺MHCII⁻ macrophages. We do not think that IFN β could be involved, as quantification of IFN β in the CHIKV-infected joint-footpad revealed no differences between control and clodronate-liposomes treated groups (see below). This further indicates that IFN β presence is not dependent on the macrophages and also IFN β do not mediate CHIKV-induced joint-footpad swelling at 6 dpi.

Figure. Quantification of IFN β in CHIKV-infected joint-footpad following clodronate liposome treatment.

Wildtype animals were infected with 1×10^6 PFU of CHIKV in the right footpad. At 5dpi, animals were given intraperitoneally with either clodronate liposome (1mg) or empty liposome (control). Levels of IFN β present in joint-footpad was quantified at 6 dpi for both groups of animals.

Nevertheless, we did not perform the experiment in IFNAR KO animals, as these mice have previously been demonstrated to be susceptible to lethal CHIKV infection (Schilte et al., *J Exp Med*, 2010).

- Schilte, Clémentine, et al. "Type I IFN controls chikungunya virus via its action on nonhematopoietic cells." *Journal of Experimental Medicine* 207.2 (2010): 429-442.

Referee #3 (Comments on Novelty/Model System for Author):

The experimental model/systems used in the manuscript are adequate and extensively used in the field.

Referee #3 (Remarks for Author):

In the manuscript Crosstalk between CD64+MHCII+ macrophages and CD4+ T cells drives joint pathology during chikungunya by Lum F-M, Chan Y-H and colleagues, authors added relevant information on immunopathogenesis of CHIKV-inflammation. Authors demonstrated that interaction between CD4+T cells and CD11b+/Ly6C+/CD64+/MHCII+ drives CHIKV disease in mice. The experiments are well design and performed and the manuscript is easy to follow. The manuscript is interesting and relevant for the field; however, some questions should be answered by the authors before publication.

1. On the figure caption EV5: check/corrected (letter D) ...CD8+ T cells, LFA-1+CD4+T cells, CD11b+Ly6C+ and ... to: ...CD8+ T cells, LFA-1+CD8+T cells, CD11b+Ly6C+CD64+MHCII-monocytes.

Response: We thank the reviewer for spotting this oversight. We have now corrected it to reflect (D)

2. Fig.1C - Authors compared using uMAP the cluster of cells between mock- and CHIKV-infected cells. Can authors comment more on cluster 9 on mock-infected animals which "disappeared or move" to another position/cluster during CHIKV-infection. Can authors phenotypically identify these cells?

Response: Based on the its low presence, yet high expression of MHCII and F4/80, we postulate that these could be resident macrophages. Upon CHIKV infection, this population of cells could have moved to another cluster, such as the cluster of interest (cluster 7), which shared the expression of numerous markers as cluster 9 as shown in Expanded View Figure 1A. Our objective of this manuscript was to identify unique myeloid populations that appear following CHIKV infection, thus the sole focus on cluster 7.

3. Authors stated (line 140-142) that cluster 7 presents high levels of CD11b and Ly6C identifying them as tissue infiltrating monocytes. However, this cluster also presents (based on figure EV1) high levels of CD3? Which proportion of the cells on this cluster are CD11b/Ly6C and CD3?

Response: Based on Expanded View Figure 1A, CD3 is co-expressed on the infiltrating CD11b+Ly6C+ cells. Based on our flow data (attached below), gating on CD11b+Ly6C+ cells revealed that approximately 50% of the population is CD3+. Further gating showed that both CD3- and CD3+ CD11b+Ly6C+ cells could further differentiate into CD64+MHCII+ phenotype. As our objective was solely on the CD11b+Ly6C+ population differentiating into the CD64+MHCII+ phenotype, we did not take into account the expression of CD3 and the effect of CD3 signalling in myeloid cells. Nevertheless, it would be interesting to study the role of CD3 macrophages in viral infections given their proinflammatory role (Rodriguez-Cruz et al., *Front Immunol*, 2019).

- Rodriguez-Cruz, Adriana, et al. "CD3+ macrophages deliver proinflammatory cytokines by a CD3-and transmembrane TNF-dependent pathway and are increased at the BCG-infection site." *Frontiers in immunology* 10 (2019): 2550.

Figure: Gating on CD11b+Ly6C+ cells.

Wildtype animals were infected with 1×10^6 PFU of CHIKV in the right footpad. Immunophenotyping was performed at 6 dpi. Plots show the gating of CD3 in CD11b+Ly6C+ cells. Further gating showed that both CD3- and CD3+ CD11b+Ly6C+ cells are capable of differentiating into the CD64+MHCII+ macrophage phenotype.

4. Authors demonstrated the role of the depletion of GM-CSF in CHIKV pathogenesis and viral clearance (Fig.3). Authors started the depletion of GM-CSF at 4 dpi and showed a reduction in joint-footpad swelling (Fig3A). However, authors indicated a reduction in viral RNA load in the first days of the infection (2-3 days) (line 207-209). Thus, this reduction in viral RNA load occurs before GM-CSF depletion which started only at 4 dpi. How authors explain this viral RNA load reduction?

Response: We want to thank the reviewer for pointing this out and we really appreciate this. Instead of comparing using the non-parametric Mann-Whitney *U* test as stated, the results were analysed with a parametric test. We have since addressed this oversight, and have gone through all the statistical analyses presented in this manuscript to ensure that there are no further statistical errors. We have since make the necessary changes to the manuscript text.

5. Suggestion: Authors demonstrated the impact of CD4 depletion on immune cells infiltrates and cytokine/chemokine secretion on footpad (Fig4C). Authors indicated a reduction on secretion of some cytokines/chemokines after CD4 depletion (lines 251-256). Specifically, authors focus on reduction of IFN γ and GM-CSF (Fig4D). However, other cytokines/chemokines are reduced after CD4 depletion and could present a role in the changes in the cell migration to the footpad of CHIKV-infected mice. Then, authors should consider to shown the reduction of other cytokine/chemokines relevant for the model in a supplementary figure.

Response: As highlighted by the reviewer, numerous cytokines were indeed affected by the lost of CD4+ T cells. As per our response to Reviewer 2, depletion of a particular immune population would cause changes in other immune populations or cytokines environment. Here, we particularly focused on GM-CSF and IFN γ as crucial cytokines in inducing the differentiation of CD64+MHCII+ macrophages, as they were demonstrated to mediate the transition of inflammatory Ly6C+ monocytes into macrophages in a murine model of neuroinflammation (Amorim et al., *Nat Immunol*, 2022). Since CD4+ T cells are capable of secreting GM-CSF and IFN γ (Campbell et al., *J Immunol*, 2011), also reported in this manuscript as Expanded View Figure 2B and 2C, thus the emphasis on these two cytokines to highlight how the depletion of CD4+ T cells affected their levels (Figure 4D). Nevertheless, we did mention in the discussion that the differentiation or activation into macrophages could also be attributed to the levels of M-CSF, which levels were also affected by the depletion of CD4+ T cells. Due to limitation on the numbers of supplementary figures we can have, we will not be able to include a figure on all the chemokines and cytokines. However, these data will be available as the source data.

- Amorim, Ana, et al. "IFN γ and GM-CSF control complementary differentiation programs in the monocyte-to-phagocyte transition during neuroinflammation." *Nature Immunology* 23.2 (2022): 217-228.
- Campbell, Ian K., et al. "Differentiation of inflammatory dendritic cells is mediated by NF- κ B1-dependent GM-CSF production in CD4 T cells." *The Journal of Immunology* 186.9 (2011): 5468-5477.

6. Authors discussed about the lack of tools to specifically depleted CD64+/MHCII+ cells. This experiment would be interest to confirm its role (together with CD4 T cells) in CHIKV-inflammation. However, would be possible to purify CD4+ T cells and CD64+/MHCII+ macrophages, and transfer those cells from CHIKV-infected mice to an uninfected mouse? The simple transfer of this cells to an uninfected mice would be sufficient to trigger inflammation and footpad swelling? I am not asking for these experiments, but I would like to listen the authors opinions on that matter, if possible.

Response: We thank the reviewer for this suggestion. However, we wish to explain that there are several factors that we would need to consider that this experiment would not bring more to the current data and conclusion:

- Firstly, we would need to perform sorting of CD4+ T cells and CD64+/MHCII+ macrophages from a large number of animals, in order to obtain enough quantity of cells for the transfer.
- Secondly, we would need to determine the amount of cells and correct ratio of CD4+ T cells to CD64+/MHCII+ for the transfer.
- Thirdly, we cannot be sure that the transferred cells will home to (or remain in) the footpad, as the milieu of cytokines to attract these cells are not present in an uninfected animal.
- Fourthly, transfer of CD4+ T cells and CD64+/MHCII+ macrophages into uninfected animals would not work, given that there is an absence of CHIKV antigen *in vivo* for continual priming and activation of the pathogenic CD4+ T cells by the CD64+/MHCII+ macrophages.

7. Finally, authors show that CD11b+/Ly6C+ are further classified as CD64+/MHCII+ or CD64+/MHCII- cells in joint footpad of CHIKV-infected mice. Also, was demonstrated that CD64+/MHCII+ present antigens to CD4+ and interact with these T cell population to drive CHIKV pathology (line 295-296). Based on the results present throughout the manuscript, like the RNA seq of both populations of monocytes/macrophages could authors commented more on potential function of monocyte/macrophage CD64+/MHCII- on the site of infection? Additionally, surprisingly to me, authors stated that only 14% of freshly isolated (from peripheral blood) CD11b+/Ly6C+ cells express MHCII (Line 187-188). MHCII is generally used as a marker of monocytes/macrophages, thus, could authors have commented more on this low expression of MHCII on this cell population?

Response: The presence of CD64+ macrophages during peaked CHIKV pathology suggests an inflammatory phenotype for these cells. However, in an earlier study, the depletion of macrophages with clodronate resulted in a reduction in joint footpad swelling coupled with a delayed viremia clearance (Gardner et al., *J virol*, 2010), suggesting that macrophages could also participate in a "protective" role during CHIKV infection. As reported here, RNAseq analyses performed with CD64+/MHCII- macrophages showed that these cells were completely "opposite" to their MHCII+ counterparts and were found to express "anti-inflammatory" genes such as the *dual specific phosphatases (Dusp) 4*, *Fibronectin (Fn) 1* and *matrix metalloproteinase (Mmp) 8*. *Dusp4* regulates the mitogen-activated kinase phosphatase (MKP-2) (Caunt et al., *FEBS J*, 2013), which is a key regulator of pro-inflammatory cytokines in macrophages (Falcicchia et al., *Int J Mol Sci*, 2020). Deficiency in

MKP-2 resulted in elevated levels of pro-arthritis cytokines and greater disease severity in a murine model of inflammatory arthritis (Schroeder et al., *RMD Open*, 2019). *Fn1* (Jablonski et al., *PloS One*, 2015) and *Mmp8* (Wen et al., *J Biol Chem*, 2015) are two genes commonly associated with the anti-inflammatory M2 macrophages. *Mmp8* has also been shown to reduce inflammation in an acute lung inflammation model (Quintero et al., *J Immunol*, 2010). We did not discuss the role of CD64+MHCII- macrophages in the manuscript due to word limit constraints.

- Gardner, Joy, et al. "Chikungunya virus arthritis in adult wild-type mice." *Journal of virology* 84.16 (2010): 8021-8032.
- Caunt, Christopher J., and Stephen M. Keyse. "Dual-specificity MAP kinase phosphatases (MKPs) Shaping the outcome of MAP kinase signalling." *The FEBS journal* 280.2 (2013): 489-504.
- Falcicchia, Chiara, et al. "Involvement of p38 MAPK in synaptic function and dysfunction." *International journal of molecular sciences* 21.16 (2020): 5624.
- Schroeder, Juliane, et al. "Novel protective role for MAP kinase phosphatase 2 in inflammatory arthritis." *RMD open* 5.1 (2019): e000711.
- Jablonski, Kyle A., et al. "Novel markers to delineate murine M1 and M2 macrophages." *PloS one* 10.12 (2015): e0145342.
- Wen, Guanmei, et al. "A novel role of matrix metalloproteinase-8 in macrophage differentiation and polarization." *Journal of Biological Chemistry* 290.31 (2015): 19158-19172.
- Quintero, Pablo A., et al. "Matrix metalloproteinase-8 inactivates macrophage inflammatory protein-1 α to reduce acute lung inflammation and injury in mice." *The journal of immunology* 184.3 (2010): 1575-1588.

Additionally, surprisingly to me, authors stated that only 14% of freshly isolated (from peripheral blood) CD11b⁺/Ly6C⁺ cells express MHCII (Line 187-188). MHCII is generally used as a marker of monocytes/macrophages, thus, could authors have commented more on this low expression of MHCII on this cell population?

Regarding the low percentage of blood monocytes exhibiting MHCII, this is likely due to the monocytes being isolated fresh from the blood of the animal, and thus is in their steady-state. Typically, classical monocytes are thought to lack MHCII (Geissmann et al., *Immunity*, 2003; Ingersoll et al., *Blood*, 2010), and after culturing *in vitro* may trigger their differentiation into macrophages thus the increased expression of MHCII (Haag et al., *Bio Protoc*, 2021).

- Geissmann, Frederic, Steffen Jung, and Dan R. Littman. "Blood monocytes consist of two principal subsets with distinct migratory properties." *Immunity* 19.1 (2003): 71-82.
- Ingersoll, Molly A., et al. "Comparison of gene expression profiles between human and mouse monocyte subsets." *Blood* 115.3 (2010): e10-e19.
- Haag, Simone M., and Aditya Murthy. "Murine monocyte and macrophage culture." *Bio-protocol* 11.6 (2021): e3928-e3928.

In a report by Jakubzick et al., (*Immunity*, 2013), only ~10% of CD115⁺CD43^{lo}Ly6C⁺ blood monocytes were shown to be positive for MHCII (population 2, indicated by red arrow).

- Jakubzick, Claudia, et al. "Minimal differentiation of classical monocytes as they survey steady-state tissues and transport antigen to lymph nodes." *Immunity* 39.3 (2013): 599-610.

Figure: Analysis of blood monocytes (Figure adapted from Jakuzick et al., *Immunity*, 2013).

(A) The gating strategy leading to generation of five gates for monocytes subsets is shown in dot plots (left) and a diagram (right). Red arrow marks the gate for Ly6C⁺MHCII⁺ cells.

(B) The frequency of blood monocyte subsets in gates 1-5 from **(A)** are plotted on a log scale. Red arrow indicates the Ly6C⁺MHCII⁺ cells.

8th Jan 2024

Dear Dr. Ng,

Thank you for submitting your revised manuscript. We have now received the reports from the referees who re-reviewed your manuscript, and as you will see below, they are supportive of publication pending minor revisions and discussion. I will therefore be able to accept your manuscript once the following points will be addressed:

1/ Referees' comments:

Please address the remaining concerns from referees #1 and #2.

2/ Manuscript text:

- Please remove the yellow highlights, and only keep in track changes mode any new modification.

- Materials and Methods:

o Cells: please provide culture conditions and indicate whether the cells were tested for mycoplasma contamination.

o Mice: please indicate the origin of the mice.

o Antibodies: please provide dilutions/concentrations.

o Primers: please provide sequences.

o Statistics: please include a statement on sample size, blinding, and inclusion/exclusion criteria.

o Please correct the checklist accordingly.

- Data Availability section: Thank you for providing a reviewer token. Please note that the datasets must be public before acceptance.

- Author contributions: CRediT has replaced the traditional author contributions section because it offers a systematic machine-readable author contributions format that allows for more effective research assessment. Please remove the Authors Contributions from the manuscript and use the free text boxes beneath each contributing author's name in our system to add specific details on the author's contribution. More information is available in our guide to authors.

- Please rename "Declarations of interest" to "Disclosure statement and competing interests" and add the sentence "Dr. Lisa Ng is a Member of the EMBO Molecular Medicine Editorial Board. This has no bearing on the editorial consideration of this article for publication."

3/ Figures and Appendix:

- Legends should be removed from the main and EV figure files.

- Appendix Table S1 should be renamed "Dataset EV1" and needs a legend added to the file in a separate tab.

- The appendix needs a table of content added on the first page, with page numbers.

- The two callouts for "Table S1" should be corrected to "Appendix Table S1". Please make sure that all figures and figure panels are referenced in the text (currently, callouts are missing for Fig 2B,C ; Fig 3E-I ; Fig 4A). There are callouts for Fig 4E-I which don't exist, and Fig 6A,B are called out before Fig 5D, please correct.

4/ Source Data:

Thank you for providing Source Data. We note that you mention the impossibility to provide images for Figure 6A in the checklist, however these data are there. Did you mean images for Figure 1A, which are missing?

5/ Checklist:

- Please fill in the section on mycoplasma contamination.

- Please check the section on microbes.

- Please complete the section on experimental study design and statistics.

6/ Every published paper now includes a 'Synopsis' to further enhance discoverability. Synopses are displayed on the journal webpage and are freely accessible to all readers. They include a short stand first (maximum of 300 characters, including space) as well as 2-5 one-sentences bullet points that summarizes the paper. Please write the bullet points to summarize the key NEW findings. They should be designed to be complementary to the abstract - i.e. not repeat the same text. We encourage inclusion of key acronyms and quantitative information (maximum of 30 words / bullet point). Please use the passive voice. Please attach these in a separate file or send them by email, we will incorporate them accordingly.

7/ As part of the EMBO Publications transparent editorial process initiative (see our Editorial at

<http://embomolmed.embopress.org/content/2/9/329>), EMBO Molecular Medicine will publish online a Review Process File (RPF) to accompany accepted manuscripts.

This file will be published in conjunction with your paper and will include the anonymous referee reports, your point-by-point response and all pertinent correspondence relating to the manuscript. Let us know whether you agree with the publication of the RPF and as here, if you want to remove or not any figures from it prior to publication.

I look forward to receiving your revised manuscript.

Yours sincerely,

Lise Roth

Lise Roth, PhD

Senior Editor

EMBO Molecular Medicine

***** Reviewer's comments *****

Referee #1 (Comments on Novelty/Model System for Author):

The murine models of CHIKV infection are established.

Referee #1 (Remarks for Author):

The authors have clarified most of my questions, but I still have a few minor questions for the authors to consider.

1. The authors explained that the animal ages for Fig.2 and EV4 were 3-4 weeks. What are the ages of mice for the other experiments? This information should be clearly indicated in the methods and figure legends. The rationale for the use of 3-4-week-old mice should be clarified.
2. The authors claimed "Serum GM CSF and IFN γ levels were already depicted in Figure 4D". However, the Fig.4.C/D legend states "joint lysate .." not serum. It is okay to focus on the immune responses in joints.
3. "The UMAP plots were normalized to 5000 cells per sample, and represent the abundance, but not the quantity of cells present in the joints". Would "proportion" (or similar words) be clearer to readers?

Referee #2 (Remarks for Author):

Authors have revised the manuscript to meet the issues raised by the reviewers. In their response, they provide new information and clarify some of the issues raised in particular those related to the lack of tools and the adoptive transfer experiment. It remains a technical limitation the fact that isotope antibody was not used as a control in the antibody-based blocking experiments. This needs to be acknowledged as a limitation in the discussion. Without this control, authors cannot rule out rigorously any effect due to the high amount of antibody used in the depletion experiments.

I urge the authors to disclose in full the panel of cytokines and chemokines.

Referee #3 (Comments on Novelty/Model System for Author):

I believe that the manuscript add relevant information about a population of macrophages (CD64+/MHCII+) involved in CHIK pathogenesis. Additionally, the experiment seems to be well performed and the model used is adequate. Regarding the medical impact, despite interesting I believe that more experiments in animal are needed as a final proof of the role of CD64+/MHCII+ macrophages as a target for CHIKV treatment.

Referee #3 (Remarks for Author):

I believe that the manuscript is suitable for publication at the EMBO Molecular Medicine as an original manuscript.

Referee #1

The murine models of CHIKV infection are established.

Referee #1 (Remarks for Author):

The authors have clarified most of my questions, but I still have a few minor questions for the authors to consider.

1. The authors explained that the animal ages for Fig.2 and EV4 were 3-4 weeks. What are the ages of mice for the other experiments? This information should be clearly indicated in the methods and figure legends. The rationale for the use of 3-4-week-old mice should be clarified.

Responses: We have indicated that all animals used in this manuscript are aged between three- to four-week old, gender-matched, C57BL/6J background. These animals were used as they would be able to give a more pronounced joint-footpad pathology.

2. The authors claimed "Serum GM CSF and IFN γ levels were already depicted in Figure 4D". However, the Fig.4.C/D legend states "joint lysate .." not serum. It is okay to focus on the immune responses in joints.

Responses: This comment is in reference to our response in the rebuttal. Yes, reviewer #1 is correct to say that it should have been joint lysate and not serum. In fact the data presented in Figure 4C and 4D were obtained from the joint lysate as stated in the figure legend.

3. "The UMAP plots were normalized to 5000 cells per sample, and represent the abundance, but not the quantity of cells present in the joints". Would "proportion" (or similar words) be clearer to readers?

Responses: We agree and have since substituted the word “abundance” with “proportion” throughout the manuscript.

Referee #2

Authors have revised the manuscript to meet the issues raised by the reviewers. In their response, they provide new information and clarify some of the issues raised in particular those related to the lack of tools and the adoptive transfer experiment. It remains a technical limitation the fact that isotope antibody was not used as a control in the antibody-based blocking experiments. This needs to be acknowledged as a limitation in the discussion. Without this control, authors cannot rule out rigorously any effect due to the high amount of antibody used in the depletion experiments.

Responses: We have now added this into the discussion as a limitation to read, “Another limitation in our study was that an isotope antibody was not utilized as a control in our antibody-based *in vivo* depletion experiments as we wanted the infection to resemble closely to that of the non-treated animals. Nevertheless, we have shown in Appendix Fig S1, disease severity and viral load clearance were comparable between animals receiving PBS and isotype control antibodies.” On lines 406-411 in the revised manuscript.

I urge the authors to disclose in full the panel of cytokines and chemokines.

Responses: As mentioned in our previous rebuttal, the entire list of cytokines or chemokines is available in the Figure Source Data. Furthermore, the entire list of cytokines or chemokines evaluated is also presented in Figure 4C.

Referee #3

I believe that the manuscript add relevant information about a population of macrophages (CD64+/MHCII+) involved in CHIK pathogenesis. Additionally, the experiment seems to be well performed and the model used is adequate. Regarding the medical impact, despite interesting I believe that more experiments in animal are needed as a final proof of the role of CD64+/MHCII+ macrophages as a target for CHIKV treatment.

Referee #3 (Remarks for Author):

I believe that the manuscript is suitable for publication at the EMBO Molecular Medicine as an original manuscript.

Responses: We thank the reviewer for this very positive support.

17th Jan 2024

Dear Dr. Ng,

Thank you for sending the revised files. I am pleased to inform you that your manuscript is accepted for publication and is now being sent to our publisher to be included in the next available issue of EMBO Molecular Medicine!

If you have any questions, please do not hesitate to contact the Editorial Office. Thank you for your contribution to EMBO Molecular Medicine!

With kind regards,

Lise Roth
